# TREE SEARCH FOR LLM AGENT REINFORCEMENT LEARNING

**Yuxiang Ji**[1,2][*], **Ziyu Ma**[2], **Yong Wang**[2][†‡], **Guanhua Chen**[3], **Xiangxiang Chu**[2], **Liaoni Wu**[1][‡]

[1]Xiamen University  [2]AMAP, Alibaba Group  [3]Southern University of Science and Technology

 GitHub: https://github.com/AMAP-ML/Tree-GRPO

## ABSTRACT

Recent advances in reinforcement learning (RL) have significantly enhanced the agentic capabilities of large language models (LLMs). In long-term and multi-turn agent tasks, existing approaches driven solely by outcome rewards often suffer from the problem of sparse supervision. To address the challenge, we propose Tree-based Group Relative Policy Optimization (Tree-GRPO), a grouped agent RL method based on tree search, where each tree node represents the complete agent interaction step. By sharing common prefixes, the tree search sampling increases the number of rollouts achievable within a fixed budget of tokens or tool calls. Moreover, we find that the tree-structured trajectory naturally allows the construction of step-wise process supervised signals even using only the outcome reward. Based on this, Tree-GRPO estimates the grouped relative advantages both on intra-tree and inter-tree levels. Through theoretical analysis, we demonstrate that the objective of intra-tree level group relative policy optimization is equivalent to that of step-level direct preference learning. Experiments across 11 datasets and 3 types of QA tasks demonstrate the superiority of the proposed tree-based RL over the chain-based RL method.

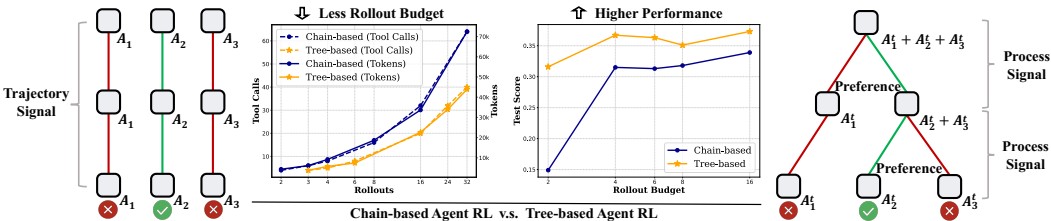

Figure 1: Comparison of chain-based and tree-based sampling strategies in LLM multi-turn agent RL. The tree structure brings two major advantages: (i) less rollout budget (both on tokens and tool-calls); (ii) higher performance.

## 1 INTRODUCTION

Reinforcement Learning (RL) has emerged as a pivotal post-training paradigm for Large Language Models (LLMs), catalyzing the development of several frontier models (DeepSeek-AI Team, 2025; Yang et al., 2025a; OpenAI, 2024). RL-tuned LLMs trained only with outcome rewards acquire complex reasoning abilities and achieve remarkable gains in single-turn tasks, such as mathematical proof and code generation (Team et al., 2025b; Yu et al., 2025; Chu et al., 2025a; Shao et al., 2024; Xin et al., 2024; Dai et al., 2026). This suggests that LLMs can learn not only through static imitation, but also by actively interacting with dynamic environments. Guided by this prospect, recent works have extended this RL paradigm to more complex agent settings involving dynamic,

---

[*]Work done during internship at AMAP, Alibaba Group.

[†]Project lead.

[‡]Corresponding author.

Figure 2: Comparison between chain-based and tree-based rollout at different levels. **Left**: Chain-based rollout. **Mid**: Tree search with nodes corresponding to tokens/sentence. **Right** (Ours): Tree search with nodes corresponding to complete agent step.

multi-turn interactions (Feng et al., 2025b; Singh et al., 2025; Ji et al., 2026; Qian et al., 2025; Feng et al., 2025a). It is believed that such agentic intelligence through long-horizon interaction in open-ended environments is essential for next-generation foundation models (Team et al., 2025a).

The agentic RL manifests in two key challenges: *i) Heavy budget taken in LLM rollouts.* Agent settings require LLMs to interact with the environment over multi-turns and complete tasks through sequential decision-making, which consequently leads to agent trajectories with thousands or more tokens alongside multiple tool-calls (Dong et al., 2025b; Feng et al., 2025b). Existing group-based RL methods sample multiple independent trajectories for each task in a chain-based rollout scheme, with considerable redundancy in the sampling process. This is reflected in practical agentic RL by the fact that the rollout phase dominates overall training time and tool-calls can incur substantial costs (e.g., high-priced search APIs). *ii) Sparse supervision in long-horizon, multi-turn trajectories.* Although agent trajectories grow with the number of turns, current agent RL approaches are still primarily driven by outcome rewards. Such trajectory-level sparse signals make it difficult to identify which specific steps or actions in a multi-turn, interdependent sequence contributed to success or failure. This means that even with a substantial increase in rollout budget, the sampled agent trajectories are still supervised by the same limited amount of training signal, resulting in a highly imbalanced learning process and even the training collapse (Wang et al., 2025c;b; Jin et al., 2025b). These two challenges raise a question: *Can we construct more fine-grained supervision signals for agent RL under a limited rollout budget while still solely based on outcome rewards?*

In this paper, we propose Tree-based Group Relative Policy Optimization (Tree-GRPO) with an online rollout strategy based on tree search. Unlike current RL approaches that independently sample complete trajectory rollouts (Figure 2 left), we replace the chain-based sampling logic with a tree-search process, which yields interleaved trajectories with shared prefix segments. Existing tree-based RL methods (Hou et al., 2025; Li et al., 2025f; Wan et al., 2024; Guo et al., 2025; Kazemnejad et al., 2024) often use token/sentence-level units as tree nodes (Figure 2 mid). For the agent tasks that have a clear step structure, it is natural to treat a complete Thought-Action-Observation step as the tree node unit (Figure 2 right). This design with clear contextual segmentation proves more suitable for agent RL (Appendix B.3) and explicitly constrains the rollout budget in both tokens and tool-calls. Under the same budget, our tree-search method can obtain around $1.5\times$ samples (depending on the tree structure) compared to the chain-based method, which is highly significant for multi-turn agentic RL training where rollout costs are substantial.

Furthermore, to address the challenge of sparse supervision, we construct more fine-grained process supervision signals by estimating relative advantages based on the tree structure. Specifically, at every branching point of the tree, we back-propagate outcome rewards from the respective subtree leaves. The differences across sibling branches serve as a preference-learning objective, providing process-level supervision signals between subtrees, where the subtree depth determines the granularity of the process signal. Since our tree search strategy uses a random expansion, it inherently yields process signals of varying granularity, enabling the model to learn intermediate decision making. This meticulous design leverages the tree structure to transform trajectory-level signals into process-level supervision. Its reliance solely on outcome rewards without additional supervision highlights its scalability and plug-and-play nature.

In our experiments, we evaluate Tree-GRPO on 11 datasets across single-hop and multi-hop knowledge-intensive tasks, along with hard web-agent tasks. Compared to chain-based methods, our proposed tree-based method demonstrates consistent improvements across models of varying series and scales. It is noteworthy that Tree-GRPO can successfully enable a base model to adopt a pre-defined multi-turn agent interaction paradigm without any supervised fine-tuning (SFT), despite

operating under an extremely limited rollout budget (tokens/tool calls). Based on Qwen2.5-3b, our Tree-GRPO achieves superior performance over the chain-based method while using only a quarter of the rollout budget. Our contributions are summarized as follows:

- We introduce a tree-based rollout strategy with nodes anchored at the agent step level in place of independent chain-based approaches for multi-turn agentic RL.
- We propose group-relative advantage estimation in both intra-tree and inter-tree level, incorporating an implicit step-level preference learning objective with a relatively stable baseline estimate.
- We provide theoretical and empirical evidence that Tree-GRPO outperforms chain-based methods in agentic RL, attaining higher performance under less rollout budgets.

## 2 PRELIMINARY

### 2.1 MULTI-TURN AGENT FRAMEWORK

We adopt the widely used ReAct (Yao et al., 2022) as the agent framework. Unlike static single-turn interaction, the agent engages in multi-turn Thought-Action-Observation cycles with the environment to solve a given task. Specifically, at each step $t = 0, 1, ..., T - 1$, the LLM generates a thought $\tau_t$ and a parsable textual action $\alpha_t$ based on the existing context $s_t$. The action typically corresponds to tool use, through which the agent dynamically interacts with the environment to obtain new observations $o_t$. A complete $T$-step agent episode consists of three interleaved trajectories:

$$\mathcal{H} = \{(\tau_0, \alpha_0, o_0), (\tau_1, \alpha_1, o_1), ..., (\tau_{T-1}, \alpha_{T-1}, o_{T-1})\}. \tag{1}$$

Such trajectories grow linearly with the number of steps, and for complex tasks requiring multiple interactions, the full trajectory can reach tens of thousands of tokens.

Following analysis in related work (Wang et al., 2025b), such a dynamic process can be described as a Markov Decision Process $\mathcal{M} = \{S, A, P\}$, where $S$ represents states (the complete interaction context up to a given time step $\mathcal{H}_{<t}$), $A$ denotes the compound action space (each action comprising a thought-action pair $(\tau_t, \alpha_t)$), and $P$ denotes the transition dynamics (includes both the external environment $P_{\text{env}}$ and the concatenation of the full context over time steps). The complete process can be formulated based on LLM policy model $\pi_\theta$ as:

$$p_\theta(s_{0:T}, \tau_{0:T}, \alpha_{0:T}, o_{0:T}) = p(s_0) \prod_{t=0}^{T-1} \left[ \pi_\theta(\tau_t|s_t) \pi_\theta(\alpha_t|s_t, \tau_t) P_{\text{env}}(o_{t+1}|\alpha_t) \right]. \tag{2}$$

### 2.2 AGENTIC REINFORCEMENT LEARNING

After formalizing the ReAct-based process as a Markov Decision Process, RL can be directly applied to optimize over the policy space by maximizing the expected return of the full state-action trajectory (Wang et al., 2025b; Dong et al., 2025b; Zhang et al., 2025b):

$$J(\theta) = \mathbb{E}_{\mathcal{H} \sim p_\theta}[R(\mathcal{H})]. \tag{3}$$

In practice, optimization is performed with a variance-reduced advantage estimator $\hat{A}(\mathcal{H})$, which stabilizes gradient updates (Schulman et al., 2015; DeepSeek-AI Team, 2025; Zhang et al., 2025a). Most existing agentic RL systems adopt an outcome-based reward, where a single scalar reward $R(\cdot)$ determined by predefined rules or model-based scoring functions is delivered to the entire trajectory.

Our method is built upon the group-based RL algorithm (DeepSeek-AI Team, 2025). Unlike estimating advantages based on extra value functions like PPO (Schulman et al., 2015), the group-based RL methods estimate advantages $\hat{A}$ by sampling a group of $N$ candidate rollouts to estimate an in-group baseline to guide the optimization direction.

## 3 TREE-BASED GROUP RELATIVE POLICY OPTIMIZATION (TREE-GRPO)

To achieve a more effective allocation of the rollout budget and address the sparse supervision challenges in multi-turn agentic RL, we propose employing tree-search-based sampling. By sharing

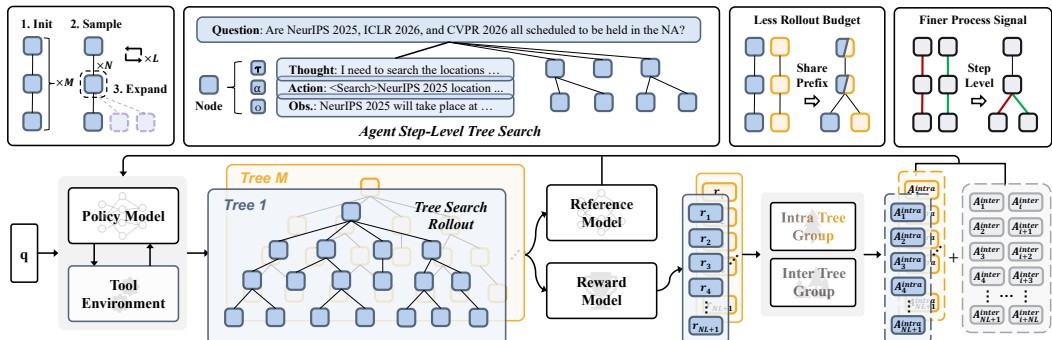

Figure 3: The overview of the Tree-GRPO training pipeline. The rollout is conducted in a tree-search manner, where each node corresponds to a complete thought-action-observation step. The group relative advantages are estimated at both intra-tree and inter-tree levels. Tree-GRPO constructs step-level process supervision signals through a tree structure with a less rollout budget.

partial prefixes between rollouts, tree search method could obtain more rollouts under the same token/tool-call budget. Based on the tree structure, we can further derive step-level process signals purely from outcome rewards, introducing an implicit step-level preference learning target into online RL. Figure 3 presents an overview of our proposed Tree-GRPO. In the following section, we will detail the implementation of agent tree search (§ 3.1), the construction of tree-structured group relative advantages (§ 3.2), and the analysis of step-level process signals (§ 3.3).

## 3.1 TREE SEARCH FOR AGENT ROLLOUT

Tree-search methods such as Monte Carlo Tree Search (MCTS) have proven effective for constructing offline DPO datasets (Xie et al., 2024; Lai et al., 2024) and for test-time scaling (Yao et al., 2023; Xin et al., 2025a), yet they are seldom used in online RL training. The main obstacle is its reliance on multi-turn sequential rollouts, which are poorly suited for parallelized LLM inference engines. This mismatch bottlenecks rollout throughput and severely limits online RL efficiency.

To address this, we adopt an initialize-then-expand approach, in which multiple chains are first initialized in parallel and nodes are then iteratively sampled for expansion. Furthermore, the clear structured agent step-level process allows us to replace the token-level nodes with agent step-level nodes, meaning that each node in the tree represents a corresponding $(\tau, \alpha, o)$ tuple. More formally, the overall tree-search sampling process is as follows:

1. **Initialization.** For each prompt $x_i$, we first generate $M$ independent chain-based trajectories $Y = \{\mathcal{H}^i \sim \pi_\theta(\cdot|x_i)\}^M$ by policy model $\pi_\theta$ as the initialization for $M$ trees $\mathcal{T}$.
2. **Sampling.** Then we randomly sample $N$ nodes $P_i = \{p_{i,j} \in \mathcal{T}_i\}^N$ except the leaf node (agent answer) from each tree $\mathcal{T}_i$ for expansion.
3. **Expansion.** For each selected node $p_{i,j}$, we take the entire context from the root to that node $\mathcal{H}^i_{<t} = \{p^{\text{root}}_{i,j}, ..., p^{\text{father}}_{i,j}, p_{i,j}\}$ and the original prompt $x_i$ as the input, continue generating the remaining part of the response by $Y_{\text{new}} = \{\mathcal{H}^i_{\geq t} \sim \pi_\theta(\cdot|x_i, \mathcal{H}^i_{<t})\}^N$, then insert it into the source tree as a new branch by $\mathcal{T}_i \leftarrow \mathcal{T}_i \cup Y_{\text{new}}$.

By iteratively repeating steps 2 and 3 $L$ times, this tree search process results in a total of $M \times (L \times N + 1)$ rollouts as final group size $G$ for a single prompt. These rollouts are evenly distributed across the $M$ trees. Let the expectation rollout budget (both in tokens and tool-calls) of a single agent trajectory be $B$. For each single random tree expansion, the expected depth of the selected node is half of the maximum depth, and the corresponding expected cost is $\frac{B}{2}$. This means we can obtain a larger number of agent rollouts for training using tree search under the same token/tool-call budget. Specifically, the total expected budget for tree-search sampling is determined by:

$$\mathbb{E}[B_{\text{tree}}] = M \cdot B + L \cdot N \cdot B/2. \tag{4}$$

Under a fixed sampling budget, decreasing tree number $M$ while increasing expansion number $N, L$ can raise the number of rollouts, but it also narrows the exploration scope, as more trajectories share the same prefix. In our experiments, different tree configurations exhibit varying effects.

## 3.2 TREE-BASED GROUP RELATIVE ADVANTAGES

Beyond enabling more rollouts under the fixed budget, a more significant potential advantage of tree search lies in the process supervision signals naturally embedded within the tree structure.

Given a group of complete trajectory rollouts $\{\mathcal{H}^i\}^G$ based on each prompt, a naive way to apply group-based policy optimization for agent RL is to organize the rollouts into trajectory-level groups. For each rollout, the reward $R(\cdot)$ is only computed at the outcome, and thus the advantage estimation is also at the trajectory level. This means that the entire multi-turn agent trajectory including multi-steps is assigned an identical credit as:

$$A(\mathcal{H}) = A(\{(\tau_0, \alpha_0, o_0), ..., (\tau_T, \alpha_T, o_T)\}) = A(\{\tau_0, \alpha_0, o_0\}) = ... = A(\{\tau_T, \alpha_T, o_T\}). \quad (5)$$

Due to the coarse credit assignment, such sparse rewards severely affect the stability of RL training for long-horizon multi-turn agents.

**Tree-based credit.** Unlike independent chain-based rollouts, tree-structured rollouts with shared prefixes naturally embed process credit signals. As shown in Figure 4, at every branching point of the tree, the difference between the back-propagated outcome rewards from respective leaves naturally constitutes a preference-learning objective for the different sub-trees. Such a form of preference learning results in process signals of varying granularity modulated by subtree depth. To achieve this form of tree-based credit assignment, we perform grouped advantage estimation within each tree $G_{\text{intra-tree}}(\mathcal{T}_i)$, serving as:

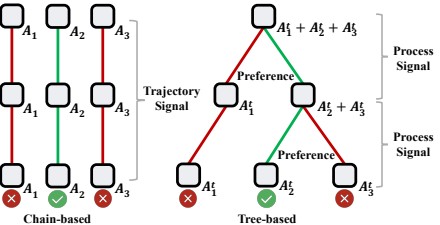

Figure 4: Comparison between chain-based and tree-based rollouts.

$$\hat{A}_{\text{Intra/Inter-tree}}(\mathcal{H}^i) = \left[R(\mathcal{H}^i) - \text{mean}(\{R(\mathcal{H}^j)\}_j^{G_{\text{Intra/Inter-tree}}(\mathcal{T}_i)})\right]/\text{std}(\{R(\mathcal{H}^j)\}_j^{G_{\text{Intra/Inter-tree}}(\mathcal{T}_i)}). \quad (6)$$

Although the intra-tree group relative advantage incorporates explicit preference objectives, the limited number of rollouts within each tree may lead to unreliable baseline estimation. To better stabilize the RL training, we also group rollouts across inter-trees (rollouts from all trees) and combine the intra-tree and inter-tree group relative advantages to obtain the final advantage estimate as:

$$\hat{A}_{\text{tree}}(\mathcal{H}^i) = \hat{A}_{\text{Intra-tree}}(\mathcal{H}^i) + \hat{A}_{\text{Inter-tree}}(\mathcal{H}^i). \quad (7)$$

The final tree-based group relative policy optimization object is:

$$J_{\text{Tree-GRPO}}(\theta) = \mathbb{E}_{x \sim \mathcal{D}, \mathcal{H} \overset{\text{tree-search}}{\sim} \pi_{\text{old}}(\cdot|x)} \left[\frac{1}{G}\sum_{i=1}^{G}\frac{1}{|\mathcal{H}^i|}\sum_{t=1}^{|\mathcal{H}^i|}\min\Big(r_{i,t}(\theta)\hat{A}_{\text{tree}}(\mathcal{H}^i),\right.$$

$$\left.\text{clip}(r_{i,t}(\theta), 1-\epsilon, 1+\epsilon)\hat{A}_{\text{tree}}(\mathcal{H}^i)\Big) - \beta\mathbb{D}_{\text{KL}}\big(\pi_\theta(\mathcal{H}|x) \| \pi_{\text{ref}}(\mathcal{H}|x)\big)\right] \quad (8)$$

where $\pi_{\text{ref}}$ and $\pi_{\text{old}}$ represent the reference LLM and previous LLM, and $\mathbb{D}_{\text{KL}}$ denotes the KL divergence. The importance sampling ratio $r_{i,t}(\theta)$ is defined at the token level $t$. The complete Tree-GRPO workflow is Algorithm 1.

## 3.3 IMPLICIT STEP-LEVEL PREFERENCE LEARNING

Leaving aside cost and scalability, a potential way to address the granularity of sparse supervision is to explicitly construct step-level DPO data with positive and negative pairs, thereby enabling preference optimization at each step. In this section, to better understand Tree-GRPO in agentic RL, we establish that intra-tree GRPO admits the same gradient structure as step-level DPO, with the only difference at the weight term.

**Assumption 3.1 (Binary Preference Setting)** *For each intermediate tree node $(x, \mathcal{H}_{<t})$, the subsequent trajectory in terms of reward falls into two categories, denoted as $\mathcal{H}_{\geq t}^{win}$ and $\mathcal{H}_{\geq t}^{loss}$, with associated rewards {1,0}. The trajectory probabilities are defined as*

$$p_\theta(\mathcal{H}_{\geq t}^{win}) = 1 - p_\theta(\mathcal{H}_{\geq t}^{loss}) = \prod_{\tau=t}^{T}\pi_\theta(\mathcal{H}_\tau^{win}|x, \mathcal{H}_{<\tau}). \quad (9)$$

Under this assumption, the step-level DPO objective can be expressed as optimizing the Bradley-Terry likelihood between the winning and losing outcomes by:

$$\nabla_\theta J_{\text{step-DPO}}(\theta) \;=\; \mathbb{E}_{(x,\mathcal{H}_{<t},\mathcal{H}_{\geq t}^{\text{win}},\mathcal{H}_{\geq t}^{\text{loss}})\sim\mathcal{D}} \Bigg[ \sigma \left( \beta \log p_\theta \left( \mathcal{H}_{\geq t}^{\text{loss}} \right) - \beta \log p_\theta \left( \mathcal{H}_{\geq t}^{\text{win}} \right) \right)$$
$$\cdot \left( \nabla_\theta \log p_\theta \left( \mathcal{H}_{\geq t}^{\text{win}} \right) - \nabla_\theta \log p_\theta \left( \mathcal{H}_{\geq t}^{\text{loss}} \right) \right) \Bigg]. \tag{10}$$

Correspondingly, the gradient of intra-tree GRPO can be derived into a combined form consisting of $\mathcal{H}_{\geq t}^{\text{win}}$ and $\mathcal{H}_{\geq t}^{\text{loss}}$ as:

$$\nabla_\theta J_{\text{Intra-tree}}(\theta) \;=\; p_\theta(\mathcal{H}_{\geq t}^{\text{win}}) \cdot p_\theta(\mathcal{H}_{\geq t}^{\text{loss}}) \cdot [\nabla_\theta \log p_\theta \left( \mathcal{H}_{\geq t}^{\text{win}} \right) - \nabla_\theta \log p_\theta \left( \mathcal{H}_{\geq t}^{\text{loss}} \right)] \tag{11}$$

**Proposition 3.1 (Structural Equivalence of Step-level DPO and Intra-tree GRPO)** *Under Assumption C.1, both step-level DPO and intra-tree GRPO admit gradient estimators of the form*

$$\nabla_\theta J_{\text{unified}}(\theta) \;=\; \underbrace{w}_{\text{Weight}} \cdot \underbrace{\left( \nabla_\theta \log p_\theta \left( \mathcal{H}_{\geq t}^{\text{win}} \right) - \nabla_\theta \log p_\theta \left( \mathcal{H}_{\geq t}^{\text{loss}} \right) \right)}_{\text{Preference Advantage Gradient}}, \tag{12}$$

*where the only difference lies in the choice of the weight term $w$.*

Proposition C.1 indicates that intra-tree GRPO can be interpreted as implicitly performing step-level preference optimization, thereby inheriting the key property of step-level DPO while operating in an online rollout setting. Detailed derivations are put in Appendix C.

## 4 EXPERIMENT

### 4.1 EXPERIMENTAL SETUP

**Datasets.** To evaluate the effectiveness of our proposed Tree-GRPO in LLM agentic RL, we conduct experiments on 11 benchmarks across three categories: (i) *Multi-Hop QA* including: HotpotQA (Yang et al., 2018), 2WikiMultiHopQA (Ho et al., 2020), Musique (Trivedi et al., 2022), and Bamboogle (Press et al., 2023); (ii) *Single-Hop QA* including: NQ (Kwiatkowski et al., 2019), TriviaQA (Joshi et al., 2017), and PopQA (Mallen et al., 2023); (iii) *Web-Agent QA* including: SimpleQA (Press et al., 2023), GAIA (Mialon et al., 2023), WebWalkerQA (Wu et al., 2025b), and BrowseComp (Wei et al., 2025).

**Baselines.** We compare the proposed Tree-GRPO against the following baseline: (i) *Direct Prompting Methods* including direct inference, ReAct agent framework (Yao et al., 2022); (ii) *Advanced RAG Method* Search-o1 (Li et al., 2025c); (iii) *RL-based Methods* including GRPO (DeepSeek-AI Team, 2025) and GSPO (Zheng et al., 2025). Our implementation is built upon the Search-R1 (Jin et al., 2025b) repository, including prompt template and agent-environment interaction.

**Experimental Setups.** For all settings, we exclusively use search engines as the designated tool. For *Multi-Hop QA* and *Single-Hop QA* settings, we use an E5-based local retrieval server (Wang et al., 2022) built on a Wikipedia dump (Karpukhin et al., 2020), while the *Web-Agent QA* setting employs a real web search API for retrieval.

**Metrics.** Exact Match (EM) is used for both the training and test score in *Single-Hop QA* and *Multi-Hop QA*. For *Web-Agent QA*, we utilize F1 scores for both training and test.

We conduct experiments using two series of models Qwen-2.5 (Base/Instruct) (Qwen et al., 2024) and Llama-3.2 (Base/Instruct) (Grattafiori et al., 2024) across different parameter scales including 1.5b, 3b, 7b, and 14b. If not specified, the default rollout budget is 4 for each prompt during training. More implementation and experiment details are in Appendix A and B.

### 4.2 MAIN RESULTS

**Multi-Hop QA.** As shown in Table 1, in a multi-hop QA setting that requires multi-turn interactions, although models within the ReAct framework can acquire information via tool calls, small models (<7b parameters) exhibit little improvement over direct inference. This indicates that prompting

Table 1: Overall performance on single-hop QA and multi-hop QA, with EM scores for each dataset. The best results are indicated in **bold**.

| | Method | Single-Hop QA | | | | Multi-Hop QA | | | | |
|---|---|---|---|---|---|---|---|---|---|---|
| | | NQ | Trivia | PopQA | Avg./$\Delta_{rel}^{\%}$ | Hotpot | 2wiki | Musiq | Bamb | Avg./$\Delta_{rel}^{\%}$ |
| **Qwen2.5-1.5b** | Direct Inference | 7.1 | 22.4 | 9.9 | 13.1 | 5.9 | 4.3 | 2.6 | 8.0 | 5.2 |
| | Search-o1 | 10.2 | 30.9 | 15.0 | 15.4 | 11.6 | 12.2 | 3.1 | 13.0 | 10.0 |
| | ReAct | 9.5 | 22.1 | 13.8 | 15.1 | 7.3 | 8.0 | 1.9 | 11.2 | 7.1 |
| | + GRPO | 39.4 | 51.0 | 39.7 | 43.4 $\Delta_{base}$ | 14.6 | 24.4 | 2.2 | 4.0 | 11.3 $\Delta_{base}$ |
| | + GSPO | 36.8 | 48.9 | 37.3 | 41.0$_{-5.5\%}$ | 15.8 | 23.7 | 2.5 | 4.8 | 11.7$_{+3.5\%}$ |
| | + Tree-GRPO | **43.6** | **57.3** | **41.6** | **47.5**$_{+9.5\%}$ | **29.5** | **26.8** | **6.6** | **13.6** | **19.1**$_{+69\%}$ |
| **Qwen2.5-3b** | Direct Inference | 10.6 | 28.8 | 10.8 | 16.7 | 14.9 | 24.4 | 2.0 | 2.4 | 10.9 |
| | Search-o1 | 15.1 | 44.3 | 13.1 | 24.2 | 18.7 | 17.6 | 5.8 | 29.6 | 17.9 |
| | ReAct | 21.1 | 43.5 | 28.3 | 31.0 | 19.2 | 19.1 | 4.8 | 20.0 | 15.8 |
| | + GRPO | 44.4 | 58.0 | 42.0 | 48.1 $\Delta_{base}$ | 39.0 | 36.3 | 15.2 | 36.8 | 31.8 $\Delta_{base}$ |
| | + GSPO | 43.0 | 58.8 | 42.5 | 48.1$_{+0.0\%}$ | 40.2 | 39.8 | 17.0 | 36.8 | 33.5$_{+5.2\%}$ |
| | + Tree-GRPO | **46.8** | **59.7** | **43.6** | **50.0**$_{+4.0\%}$ | **42.4** | **43.7** | **17.8** | **43.2** | **36.8**$_{+16\%}$ |
| **Llama3.2-3b** | Direct Inference | 16.2 | 29.6 | 7.4 | 17.7 | 12.6 | 9.2 | 2.0 | 8.0 | 8.0 |
| | Search-o1 | 24.2 | 48.4 | 8.8 | 27.1 | 19.4 | 17.4 | 6.0 | 32.0 | 14.1 |
| | ReAct | 23.9 | 42.4 | 21.7 | 29.3 | 16.2 | 10.4 | 3.5 | 23.2 | 13.3 |
| | + GRPO | 45.5 | 58.2 | **42.4** | 48.7 $\Delta_{base}$ | 36.0 | 26.9 | 11.8 | 32.0 | 26.7 $\Delta_{base}$ |
| | + GSPO | 41.2 | 57.8 | 40.8 | 46.6$_{-4.3\%}$ | 28.1 | 24.5 | 8.6 | 32.0 | 23.3$_{-13\%}$ |
| | + Tree-GRPO | **47.7** | **59.9** | 42.3 | **50.0**$_{+2.7\%}$ | **44.6** | **38.4** | **17.6** | **46.4** | **36.8**$_{+38\%}$ |
| **Qwen2.5-7b** | Direct Inference | 13.4 | 40.8 | 14.0 | 22.7 | 18.3 | 25.0 | 3.1 | 12.0 | 14.6 |
| | Search-o1 | 23.8 | 47.2 | 26.2 | 32.4 | 22.1 | 21.8 | 5.4 | 32.0 | 20.3 |
| | ReAct | 30.6 | 56.3 | 34.6 | 40.5 | 27.9 | 25.3 | 11.3 | 28.8 | 23.3 |
| | + GRPO | 45.8 | 61.5 | 44.3 | 50.5 $\Delta_{base}$ | 42.5 | 40.7 | 19.1 | 43.2 | 36.4 $\Delta_{base}$ |
| | + GSPO | 47.0 | **64.5** | **46.1** | **52.5**$_{+4.0\%}$ | 40.0 | 38.2 | 19.2 | **44.0** | 35.4$_{-2.8\%}$ |
| | + Tree-GRPO | **48.1** | 63.3 | 45.2 | 52.2$_{+3.4\%}$ | **44.6** | **42.3** | **20.2** | **44.0** | **37.8**$_{+3.9\%}$ |
| **Qwen2.5-14b** | Direct Inference | 19.8 | 53.1 | 18.4 | 30.4 | 21.7 | 25.3 | 4.5 | 16.0 | 16.9 |
| | Search-o1 | 34.7 | 63.5 | 24.1 | 40.8 | 26.8 | 16.1 | 9.9 | 41.6 | 23.6 |
| | ReAct | 36.1 | 64.2 | 39.3 | 46.5 | 39.1 | 33.8 | 15.0 | 43.2 | 32.8 |
| | + GRPO | 51.3 | 67.2 | 46.7 | 55.1 $\Delta_{base}$ | 47.7 | 42.6 | 23.2 | 53.6 | 41.8 $\Delta_{base}$ |
| | + GSPO | 50.7 | 67.4 | 47.1 | 55.1$_{+0.0\%}$ | 50.1 | 50.2 | 23.8 | 52.8 | 44.2$_{+5.7\%}$ |
| | + Tree-GRPO | **51.7** | **68.1** | **47.3** | **55.7**$_{+1.1\%}$ | **50.2** | **50.5** | **25.9** | **54.4** | **45.3**$_{+8.4\%}$ |

alone is insufficient for models to complete long-horizon agent tasks. Among RL approaches, our Tree-GRPO method achieves a substantial margin over chain-based GRPO baseline on models below 3b, yielding relative improvements ranging from 16% to 69% across both the Llama and Qwen series models. Remarkably, Tree-GRPO remains effective on Qwen2.5-1.5b, whereas chain-based methods struggle to stimulate multi-turn tool-use behavior. Although RL offers limited benefits on Qwen2.5-14b, our tree-based method still achieves an average relative improvement of 8.4%. These results demonstrate the superiority of the process signal provided by the tree-based method.

**Single-Hop QA.** In the single-hop QA setting which requires fewer interaction turns, the 14b model already exhibits agentic capabilities to complete tasks under the ReAct framework as Table 1. Compared to chain-based RL methods, Tree-GRPO still shows stable improvements, especially for small models like Qwen2.5-1.5b and Qwen2.5-3b. However, for most single-hop questions, the agent does not require multi-turn ReAct-style interactions and can usually solve the problem with just one round of retrieval followed by one round of answering. Due to the tree depth in this setting being limited (typically 2), the gains from process-level signals over trajectory-level are also limited.

**Web-Agent QA.** Existing open-source web-agent QA benchmarks are predominantly test sets, with a notable lack of training sets. Moreover, most of these test benchmarks are highly challenging, with some tasks requiring dozens of web interactions. The limited training data we were able to collect fails to match this level of difficulty and quality. Consequently, the performance improvement from RL is relatively limited, as shown in Table 2. In this case, Tree-GRPO consistently outperforms the chain-based GRPO across four test datasets, most notably on GAIA with a 28% average improvement. However, on more challenging benchmarks such as BrowseComp, RL yields only marginal gains, which is primarily constrained by the training data.

Table 2: Overall performance on web-agent QA, with F1 scores for each dataset. The best results are indicated in **bold**.

| | Method | SimpleQA | General AI Assistant | | | | WebWalkerQA | | | | BrowseComp |
|---|---|---|---|---|---|---|---|---|---|---|---|
| | | Avg. | Lv.1 | Lv.2 | Lv.3 | Avg. | Easy | Med. | Hard | Avg. | Avg. |
| Direct 32b | Qwen2.5-32b-Instruct | 7.7 | 8.8 | 7.7 | 3.0 | 7.6 | 6.2 | 9.4 | 5.8 | 7.4 | 2.2 |
| | DeepSeek-R1-Distill-32b | 12.6 | 19.2 | 7.8 | 4.1 | 11.7 | 9.4 | 13.3 | 9.4 | 11.0 | 2.4 |
| Qwen2.5 7b | ReAct | 25.1 | 6.2 | 3.5 | 1.1 | 4.2 | 8.0 | 9.2 | 5.6 | 7.6 | 1.3 |
| | + GRPO | 61.5 | 17.7 | 14.9 | 4.5 | 14.7 | 8.9 | 11.4 | 11.6 | 10.9 | 2.3 |
| | + Tree-GRPO | **62.4** | **19.3** | **17.5** | **5.7** | **16.8** | **9.3** | **11.8** | **11.9** | **11.2** | **2.7** |
| Qwen2.5 14b | ReAct | 43.3 | 11.4 | 7.1 | 0.9 | 8.0 | 9.5 | 11.3 | 7.4 | 9.5 | 1.2 |
| | + GRPO | 65.4 | **21.6** | 15.0 | 5.5 | 16.4 | **11.4** | 14.8 | 10.3 | 12.4 | 2.4 |
| | + Tree-GRPO | **67.8** | 20.8 | **24.3** | **7.3** | **21.0** | 11.1 | **15.5** | **10.8** | **12.8** | **2.6** |

Table 3: Performance with different training budget (defined as the cost of several complete agent trajectories per prompt). The base model is Qwen2.5-3b. The best results are indicated in **bold**.

| Method | Single-Hop QA | | | | Multi-Hop QA | | | | |
|---|---|---|---|---|---|---|---|---|---|
| | NQ | Trivia | PopQA | Avg./$\Delta_{rel}^{\%}$ | Hotpot | 2wiki | Musiq | Bamb | Avg./$\Delta_{rel}^{\%}$ |
| *Rollout Token/Tool Budget $\approx$ 2/per prompt* | | | | | | | | | |
| Chain-based | 42.0 | 56.7 | 40.8 | 46.5 $\Delta_{base}$ | 17.9 | 25.6 | 3.3 | 12.8 | 14.9 $\Delta_{base}$ |
| Tree-based ($M=1, N=2, L=1$) | **46.1** | **59.4** | **43.6** | **49.7**$_{+6.9\%}$ | **39.5** | **40.2** | **13.7** | **32.8** | **31.6**$_{+112\%}$ |
| *Rollout Token/Tool Budget $\approx$ 4/per prompt* | | | | | | | | | |
| Chain-based | 44.4 | 58.0 | 42.0 | 48.1 $\Delta_{base}$ | 39.0 | 36.3 | 15.2 | 36.8 | 31.8 $\Delta_{base}$ |
| Tree-based ($M=2, N=2, L=1$) | **46.8** | **59.7** | **43.6** | **50.0**$_{+4.0\%}$ | **42.4** | **43.7** | **17.8** | **43.2** | **36.8**$_{+16\%}$ |
| *Rollout Token/Tool Budget $\approx$ 8/per prompt* | | | | | | | | | |
| Chain-based | 46.5 | 59.2 | 44.3 | 50.0 $\Delta_{base}$ | 39.4 | 36.4 | 16.1 | 33.6 | 31.4 $\Delta_{base}$ |
| Tree-based ($M=4, N=2, L=1$) | **47.6** | **60.8** | 44.2 | **50.8**$_{+1.6\%}$ | 42.0 | **42.9** | **19.5** | 36.0 | 35.1$_{+12\%}$ |
| Tree-based ($M=2, N=6, L=1$) | 46.9 | 59.7 | **44.5** | 50.4$_{+0.8\%}$ | **42.2** | 42.6 | 18.3 | **42.4** | **36.4**$_{+16\%}$ |
| *Rollout Token/Tool Budget $\approx$ 16/per prompt* | | | | | | | | | |
| Chain-based | 47.8 | 61.1 | 44.7 | 51.2 $\Delta_{base}$ | 40.1 | 38.8 | 17.5 | 39.2 | 33.9 $\Delta_{base}$ |
| Tree-based ($M=8, N=2, L=1$) | **48.6** | **61.7** | 44.9 | **51.7**$_{+1.0\%}$ | 44.6 | 43.2 | 18.2 | 38.4 | 36.1$_{+6.5\%}$ |
| Tree-based ($M=6, N=3, L=1$) | 48.5 | 61.6 | **45.0** | **51.7**$_{+1.0\%}$ | **45.3** | **44.1** | 18.8 | 37.6 | 36.5$_{+7.7\%}$ |
| Tree-based ($M=4, N=5, L=1$) | 48.4 | 61.3 | 43.8 | 51.2$_{+0.0\%}$ | 45.0 | 43.9 | 18.5 | **41.6** | **37.3**$_{+10\%}$ |

## 4.3 QUANTITATIVE ANALYSIS

In this section, we extend our study to more training configurations and analyze what the tree-based method affords beyond performance.

**Different Training Budget.** In LLM agent RL training, the token/tool-call costs introduced by multi-turn interactions are an important concern. Here we assess our method under different cost constraints. As shown in Table 3, the tree-based method consistently demonstrates improvements under different budget settings. Especially under highly constrained rollout budgets (e.g., when only two complete rollouts budget per prompt), chain-based RL struggles to learn multi-turn interactions, whereas the tree-based method achieves substantially better results (112% relative improvement). As the rollout budget increases, the superiority of the tree-based method having more training trajectories gradually diminishes in the single-hop setting, whereas the benefit of finer process supervision signals remains in the multi-hop setting. *Remarkably, our Tree-GRPO achieves superior performance over the chain-based method while using only a quarter of the rollout budget.* In addition, when the rollout budget is larger, tree-based sampling offers more flexibility in parameter choices. More analysis is in Appendix B.4.

**Chain-based vs. Tree-based Beyond Performance.** Due to the sparse nature of outcome rewards in multi-turn agentic RL, the model often struggles to learn more complex processes, tending instead to favor shorter interaction paths rather than extended exploration, and in some cases even learning toward unreasonable shortcuts. While such behavior is generally acceptable, it becomes a limitation for agent tasks that inherently require longer multi-turn interactions. In our experiments on the multi-hop QA setting in Figure 5, we find that beyond performance improvements in training reward, the tree-based method also *encourages the LLM agent to engage in longer interactions* (i.e.,

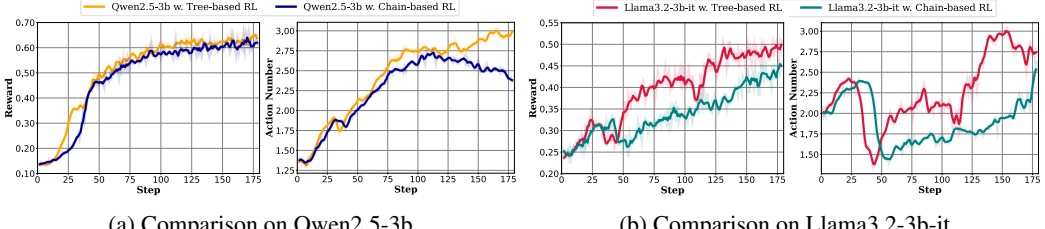

Figure 5: Comparison between tree-based and chain-based RL on reward and action number.

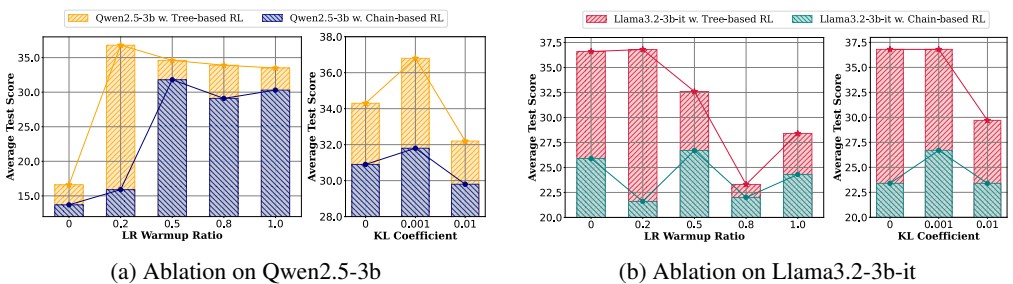

Figure 6: Ablation study on learning rate warmup ratio and KL term coefficient.

making more tool calls) from 2.4 to 3.0 on average to solve each multi-hop QA. This is particularly meaningful for training agents capable of solving more complex long-horizon tasks. Moreover, we observe that the learning rate (LR) warmup is a particularly sensitive hyperparameter when training small models below 3b. The results in Figure 6 show that our tree-based method outperforms the chain-based method under all LR warmup ratio settings. In addition, we conduct an ablation study on the coefficient of the KL term, which is often crucial in LLM RL. The results also highlight that the tree-based method maintains stability across different settings, and that an appropriately weighted KL term can help the model learn better within a constrained exploration space.

**Tree-based Advantage.** In Table 4, we compare the performance of different advantage estimation methods. When the rollout budget is relatively limited (e.g., $M = 2, N = 2, L = 1$), the number of branches within a single tree is small. This scarcity of intra-tree grouped responses results in high-variance $\hat{A}_{\text{Intra-tree}}$ and unstable baseline estimation, making RL training highly prone to collapse. This issue is absent when the intra-tree branch count is sufficient (e.g., $N = 6$). Leveraging the symmetric step-level preference learning property of $\hat{A}_{\text{Intra-tree}}$, the $\hat{A}_{\text{Inter-tree}}$ can act as a fallback baseline estimator when intra-tree branching is insufficient. In our experiments, combining $\hat{A}_{\text{Intra-tree}}$ and $\hat{A}_{\text{Inter-tree}}$ yields more stable and improved training performance across various $NML$ settings. We argue that this approach enables agent RL to incorporate the step-level preference learning property while preserving greater stability. It is worth noting that due to the more efficient sampling of tree search under the same budget, when using vanilla global group relative advantage estimation (e.g., $\hat{A}_{\text{Inter-tree}}$), the tree-based results are still better than the chain-based GRPO.

## 5 RELATED WORK

**Reinforcement Learning for Sparse Rewards.** Existing sparse-reward RL methods can generally be divided into three categories. (i) Exploration-oriented methods mitigate sparse rewards by encouraging the model to visit potentially rewarding states via curiosity (Pathak et al., 2017), goal-conditioned replay (Andrychowicz et al., 2017), etc. Due to the strong prior knowledge from base LLM models and the highly structured agent action space, exploration is not a primary bottleneck in current LLM agent RL. (ii) Reward-shaping methods (Ng et al., 1999; Arjona-Medina et al., 2019) provide additional intermediate supervision signals by decomposing or predicting process feedback. Recent studies in LLM RL primarily utilize Process Reward Models (PRMs) to generate step- or token-level feedback. However, due to the high cost of step-level human annotations and the challenges in training PRMs, such approaches have yet to scale to general scenarios. (iii) Credit assignment methods (Hung et al., 2019) redistribute outcome rewards to earlier steps through heuris-

Table 4: Ablation study on tree-based advantages.

| Advantage | Hotpot | 2Wiki | Musiq | Bamb | Avg. | $\Delta_{\text{rel}}^{\%}$ |
|---|---|---|---|---|---|---|
| *Rollout Token/Tool Budget $\approx$ 4/per prompt* | | | | | | |
| Chain-based | 39.0 | 36.3 | 15.2 | 36.8 | 31.8 | $\Delta_{\text{base}}$ |
| $\hat{A}_{\text{Intra-tree}}$ ($M=2, N=2, L=1$) | collapse | collapse | collapse | collapse | collapse | - |
| $\hat{A}_{\text{Inter-tree}}$ ($M=2, N=2, L=1$) | 40.6 | 41.3 | 16.5 | 36.8 | 33.8 | +6.3% |
| $\hat{A}_{\text{Intra-tree}} + \hat{A}_{\text{Inter-tree}}$ ($M=2, N=2, L=1$) | **42.4** | **43.7** | **17.8** | **43.2** | **36.8** | +16% |
| *Rollout Token/Tool Budget $\approx$ 8/per prompt* | | | | | | |
| Chain-based | 39.4 | 36.4 | 16.1 | 33.6 | 31.4 | $\Delta_{\text{base}}$ |
| $\hat{A}_{\text{Intra-tree}}$ ($M=2, N=6, L=1$) | 40.9 | 39.7 | 17.8 | 40.0 | 34.6 | +10% |
| $\hat{A}_{\text{Inter-tree}}$ ($M=2, N=6, L=1$) | 41.1 | 40.3 | 17.9 | 34.4 | 33.4 | +6.4% |
| $\hat{A}_{\text{Intra-tree}} + \hat{A}_{\text{Inter-tree}}$ ($M=2, N=6, L=1$) | **42.2** | **42.6** | **18.3** | **42.4** | **36.4** | +16% |

tic mechanisms. A representative line of work focuses on improving credit assignment by utilizing heuristic structures (Wang et al., 2024) or sampling-based (Kazemnejad et al., 2024) techniques.

**Reinforcement Learning for LLM and Agent.** Recent advances in RL (Kaufmann et al., 2023; Lambert et al., 2024) have demonstrated the effectiveness in LLM alignment and reasoning. Although process reward models have been shown to provide more gains on complex reasoning tasks (Lightman et al., 2023; Shao et al., 2024; Zhang et al., 2025c; Wang et al., 2025c; Liu et al., 2025; Cui et al., 2025; Choudhury, 2025; Setlur et al., 2024; Li et al., 2025b), most existing works still rely solely on outcome rewards for training due to the additional costs and limited scalability of PRMs. Techniques ranging from PPO (Schulman et al., 2015), GRPO (DeepSeek-AI Team, 2025), GSPO (Zheng et al., 2025), along with more policy variants such as DAPO (Yu et al., 2025) and GPG (Chu et al., 2025b) are employed for LLM RL training. Recent works (Wang et al., 2025b; Feng et al., 2025b; Dong et al., 2025b; Xue et al., 2025; Wang et al., 2025c; Zhou et al., 2025; Wang et al., 2025a; Li et al., 2025e; Sun et al., 2025; Dong et al., 2025a; Li et al., 2025d) apply this paradigm to end-to-end agent training. In addition to these online RL approaches, another line of work (Wang et al., 2025c; Xie et al., 2024; Xiong et al., 2025; Lai et al., 2024) directly constructs step-level DPO training data in an offline manner to achieve more fine-grained optimization objectives, while increasing the complexity of the training pipeline.

**Tree Search for LLM Reasoning.** One line of LLM tree-search research focuses on test-time scaling. Yao et al. (2023); Long (2023); Snell et al. (2024); Koh et al. (2024); Zhou et al. (2024) propose tree-of-thought to allow LLMs to consider multiple reasoning paths during solving complex tasks. Xin et al. (2024; 2025a;b) employs the Monte-Carlo tree search strategy to generate diverse proof paths for theorem proving problems. Another line of research (He et al., 2024; Wan et al., 2024; Wu et al., 2024; Xie et al., 2024; Zhang et al., 2025c; Lai et al., 2024; He et al., 2024; Li et al., 2025a) is to utilize tree-search structures for constructing step-level preference learning data, which are then used in DPO or SFT. There are also some works (Hou et al., 2025; Zhang et al., 2024; Yang et al., 2025b) similar to ours that employ tree search for sampling in LLM online RL. In particular, VinePPO (Kazemnejad et al., 2024) uses Monte Carlo estimation instead of the critic model in PPO to obtain accurate value estimates. SPO (Guo et al., 2025) partitions the trajectory into segments and performs advantage estimation through tree-based segments, thereby enabling finer segment-level credit assignment. However, since these tree-related methods are still conducted at the token/sentence level and cannot be directly employed on agent tasks.

# 6 CONCLUSION

In this work, we propose Tree-based Group Relative Policy Optimization (Tree-GRPO), adopting a tree-search rollout strategy in place of independent chain-based rollouts for LLM agent RL. Based on agent step-level nodes, Tree-GRPO carries out rollout sampling over a semantically well-defined search tree. By sharing common prefixes, the tree search sampling significantly reduces the rollout budget in terms of both tokens and tool calls during training. Tree-GRPO leverages the tree structure to conduct tree-based grouping for advantage estimation, introducing an implicit step-level preference-learning objective. Empirical evaluations on 11 datasets demonstrate the superiority of our tree-based approach for agentic RL.

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

# APPENDIX

## A  DATASETS

Since the agent exhibits varying characteristics across different benchmark settings, we consider three experimental settings, where distinct training sets are employed.

### A.1  SINGLE-HOP QUESTION ANSWERING

Single-hop QA is question answering solvable with a single supporting passage. For single-hop QA training, we directly use the training split from **NQ** (Kwiatkowski et al., 2019). The whole training data includes 79,168 QA pairs.

For single-hop QA test, we utilize the following datasets:

- **Natural Questions** (NQ) (Kwiatkowski et al., 2019) test set includes 3,610 QA pairs.
- **TriviaQA** (Joshi et al., 2017) test set includes 11,313 QA pairs.
- **PopQA** (Mallen et al., 2023) test set includes 14,267 QA pairs.

### A.2  MULTI-HOP QUESTION ANSWERING

Multi-hop QA is question answering that requires compositional reasoning and aggregation of evidence across multiple documents or entities. For multi-hop QA training, we directly use the training split from **HotpotQA** (Yang et al., 2018). The whole training data includes 90,447 multi-hop QA pairs.

For multi-hop QA test, we utilize the following datasets:

- **HotpotQA** (Yang et al., 2018) test set includes 7,405 QA pairs.
- **2WikiMultiHopQA** (Ho et al., 2020) test set includes 12,576 QA pairs.
- **Musique** (Trivedi et al., 2022) is a challenging multi-hop benchmark, which requires not only the model's information retrieval capabilities, but also deeper semantic understanding and logical reasoning. The test set includes 2,417 QA pairs.
- **Bamboogle** (Press et al., 2023) is a 2-hop hand maded dataset. The test set includes only 125 QA pairs.

### A.3  WEB-AGENT QUESTION ANSWERING

Web-agent QA is question answering in interactive web environments where the agent must search, navigate, and perform single- or multi-turn exploration and information integration. The dataset for web-agent QA training consists of two parts. For the first part, we sample 2k samples from ASearcher-35K (Gao et al., 2025), where each question and answer is filtered and verified. The second part is from released samples from WebDancer (Wu et al., 2025a), which includes 200 hard

web QA pairs. Due to the high cost of real search APIs and the difficulty of obtaining high-quality training samples, the scale of our Web-Agent QA experimental setup is relatively small.

For web-agent QA test, we utilize the following datasets:

- **SimpleQA** (Press et al., 2023) includes 500 short, fact-seeking QA pairs, which is adversarially collected against GPT-4 responses.
- **GAIA** (Mialon et al., 2023) is a hard general AI assistant benchmark consists of real-world questions. It requires the model to have the abilities such as reasoning, multi-modality handling, web browsing, and generally tool-use. We only use 103 text-only questions following previous works.
- **WebWalkerQA** (Wu et al., 2025b) contains 680 web traversal QA tasks, which is splited into three levels: easy, medium and hard.
- **BrowseComp** (Wei et al., 2025) is a challenging benchmark for measuring the abilityf ro agents to browse the web. It comprises 1,266 extremely complex, hard-to-find information questions.

## B EXPERIMENT DETAILS

### B.1 IMPLEMENTATION DETAILS

Our implementation is built upon Search-R1 (Jin et al., 2025b) based on VeRL. As Table 5, for all experimental settings, we use the learning rate 1e-6, and K3 KL in loss with 0.001 coefficient. If not specified, we use group size 4 for all chain-based RL, and $(M = 2, N = 2, L = 1)$ for Tree-GRPO. In particular, we follow Jin et al. (2025a) to add a format score $\lambda_f$ to the training reward $r(y)$ as:

$$r(y) = \begin{cases} \text{score}(y) - \lambda_f & \text{if } f_{\text{format}}(y) = \text{False} \\ \text{score}(y) & \text{else} \end{cases} \tag{13}$$

where $\lambda_f$ is set to 0.2 in all experiments. The following are the different setups for different experimental settings:

- For **Single-Hop QA** and **Multi-Hop QA** settings, our standard setup includes a total training step of 180, training batch size 512, PPO mini batch size 64. The max response length is set to 4096 tokens, and the top 3 passages from local retrieval server will be passed to the agent.
- For **Web-Agent QA** setting, we set the total training setp 34 (which corresponds to 2 epoch), training batch size 128, PPO mini batch size 64. The max response length is set to 8000 tokens, and the top 10 passages from web serp API will be passed to agent.

Table 5: Hyperparameters for Tree-GRPO and baseline methods for all experiments.

| Config | Single-Hop QA | Multi-Hop QA | Web-Agent QA |
|---|---|---|---|
| optimizer | AdamW | AdamW | AdamW |
| learning rate | 1e-6 | 1e-6 | 1e-6 |
| learning rate warmup ratio | 0.285/0.5 | 0.285/0.5 | 0 |
| KL type | K3 | K3 | K3 |
| KL coefficient | 0.001 | 0.001 | 0.001 |
| training data | 79,168 | 90,447 | 2,200 |
| total training steps | 180 | 180 | 34 |
| training batch size | 512 | 512 | 128 |
| PPO mini batch size | 64 | 64 | 64 |
| max response length | 4096 | 4096 | 8000 |
| max observation length | 500 | 500 | 1000 |
| max tool-calls | 3 | 3 | 5 |
| reward metrics | EM | EM | F1 score |
| format scores | 0.2 | 0.2 | 0.2 |
| retriever | local wiki | local wiki | Bing API |
| top-K retrieval passages | 3 | 3 | 10 |

Following Jin et al. (2025b), content enclosed within <search> </search> tags is parsed as the search query, which corresponds to the Action $\alpha$. The returned search results are then wrapped in <information> </information> tags to form the Observation $o$, thereby completing the ReAct tuple $(\tau, \alpha, o)$. The prompt template for each question as:

---

**Prompt Template**

Answer the given question. You must conduct reasoning inside <think> and </think> first every time you get new information. After reasoning, if you find you lack some knowledge, you can call a search engine by <search> query </search> and it will return the top searched results between <information> and </information>. You can search as many times as your want. If you find no further external knowledge needed, you can directly provide the answer inside <answer> and </answer>, without detailed illustrations. For example, <answer> Beijing </answer>. Question:

---

## B.2 BASELINES

**Direct Inference.** For direct inference, we directly employ instruct model to answer the question. The prompt template no longer includes any tool-use instructions, and keeps only the directive "Answer the given question. You must put the answer inside <answer> and </answer>".

**Search-o1** (Li et al., 2025c) is the search-enhanced reasoning framework, which integrates the agentic RAG mechanism and reason-in-document module.

**ReAct** (Yao et al., 2022) interleaves reasoning traces "Thought" with "Actions" (tool calls) to enable deliberate, step-by-step problem solving. The model decides when to think and when to act, using observations to refine subsequent reasoning. We use the instruct model based on ReAct as the baseline. All the RL-based methods are also based on ReAct.

**GRPO** (DeepSeek-AI Team, 2025) is a group-relative policy optimization method that updates the policy using relative advantages computed across multiple trajectories for the same prompt. Compared to PPO, it discards the value/critic model and associated losses, yielding a policy-only objective that simplifies the training pipeline with fewer components and hyperparameters. Our approach is built on GRPO, and we adopt GRPO as the primary baseline for chain-based RL.

**GSPO** (Zheng et al., 2025) is a variant of GRPO that replaces token-level importance ratio calculation with trajectory-level reweighting. By unifying importance ratio computation and advantage estimation at the trajectory level, GSPO improves the stability of LLM RL training.

## B.3 TREE SEARCH AT DIFFERENT LEVELS

To verify the effectiveness of tree search at different levels, we also conduct tree search at the token/sentence level. Since existing tree-based RL methods (Hou et al., 2025; Yang et al., 2025b; Li et al., 2025f; Guo et al., 2025; Kazemnejad et al., 2024) are not designed for agent tasks and cannot be directly applied, here we separately implement a token/sentence level tree search for agent RL. Specifically, we build the tree where each node corresponds to a token, and modify step 2 of Tree-GRPO to randomly sample tokens. During sampling, we mask out tokens from the observation $o$ that are not generated by the LLM for each trajectory $\mathcal{H}^i$ in order to prevent information confusion as:

$$[\text{MASK}]_{i,j} = \begin{cases} 1, & \mathcal{H}_j^i \in \{\tau, \alpha\} \\ 0, & \mathcal{H}_j^i \in o \end{cases} \tag{14}$$

$$P_i = \text{Sample}(\{p_{i,j} \in \mathcal{T}_i \mid [\text{MASK}]_{i,j} = 1\}). \tag{15}$$

The other settings remain consistent with Tree-GRPO.

Table 6 and Figure 7 show that tree search at the token/sentence level performs worse than the agent-step level in both single-hop QA and multi-hop QA, and even falls below the performance of chain-based GRPO. We attribute this effect to two factors. On the one hand, exploring branches in the middle of an agent step lacks a clear purpose and may lead to rollout budget waste. On the other hand, the credit derived from token/sentence-level tree structure undermines the integrity of

Table 6: Test score comparison between tree search at token/sentence-level and agent step-level. The base model is Qwen2.5-3b. The rollout budget is 4/per prompt. Tree search parameters are $M = 2, N = 2, L = 1$.

| Method | Single-Hop QA | | | | Multi-Hop QA | | | | |
|---|---|---|---|---|---|---|---|---|---|
| | NQ | Trivia | PopQA | Avg. | Hotpot | 2wiki | Musiq | Bamb | Avg. |
| GRPO | 44.4 | 58.0 | 42.0 | 48.1 | 39.0 | 36.3 | 15.2 | 36.8 | 31.8 |
| Token/sentence level | 42.1 | 56.0 | 40.6 | 46.2 | 32.0 | 30.8 | 8.4 | 17.6 | 22.2 |
| Agent step level | 46.8 | 59.7 | 43.6 | 50.0 | 42.4 | 43.7 | 17.8 | 43.2 | 36.8 |

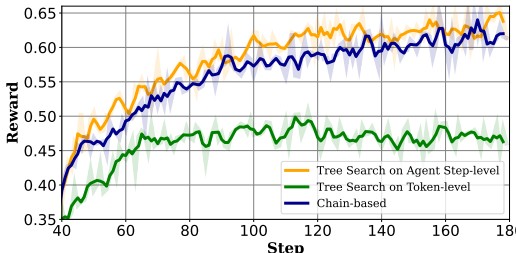

Figure 7: Training reward comparison on multi-hop QA between tree search at token/sentence-level and agent step-level.

the agent step, and the resulting process signal instead hinders the learning performance. *Therefore, we argue that tree search conducted at the token or sentence level is not suitable for agent RL tasks.*

### B.4 PERFORMANCE WITH DIFFERENT TREE STRUCTURES

We try different tree-search structures in Table 7. When $(N = 0, L = 0)$, the Tree-GRPO degenerates into vanilla GRPO. The results show that the impact of $L$ and $N$ is not much different. However, since the expansion iterations $L$ directly affect the rollout efficiency (as the iterations are executed serially), we consider using a larger $N$ and set $L = 1$ in most of the experiments. In addition, while reducing the number of trees $M$ saves rollout budget, the more constrained exploration in tree search adversely impacts the performance of RL.

Based on the results from Table 3 and Table 7, we attribute that $M$ governs the diversity of sampling (as exploration), whereas $N$ and $L$ govern the granularity of the process signal (as exploitation). The granularity of process signals required for RL training is strongly correlated with the interaction length of the task, since the tree depth is directly dictated by the number of agent steps (e.g., 3-4 steps for Multi-hop QA). In this case, when $N$ is of the same order of magnitude as or slightly larger than the tree depth, it is theoretically sufficient to construct the necessary granularity of the process signals. Given a certain rollout budget, larger values of $N, L$ (much larger than tree depth) yield diminishing marginal benefits, while a relatively smaller $M$ limits sampling diversity and results in performance degradation. Balancing $M$ and $N \times L$ can achieve a trade-off between exploration and exploitation in tree search, leading to better performance.

## C THEORETICAL ANALYSIS

**Assumption C.1 (Binary Preference Setting)** *For each intermediate tree node $(x, \mathcal{H}_{<t})$, the subsequent trajectory in terms of reward falls into two categories, denoted as $\mathcal{H}_{\geq t}^{win}$ and $\mathcal{H}_{\geq t}^{loss}$, with associated rewards {1,0}. The trajectory probabilities are defined as*

$$p_\theta(\mathcal{H}_{\geq t}^{win}) = 1 - p_\theta(\mathcal{H}_{\geq t}^{loss}) = \prod_{\tau=t}^{T} \pi_\theta(\mathcal{H}_\tau^{win}|x, \mathcal{H}_{<\tau}). \tag{16}$$

Table 7: Performance on multi-hop QA with different tree structures. The base model is Qwen2.5-3b. The best results are indicated in **bold**.

| M, N, L | Hotpot | 2wiki | Musiq | Bamb | Avg. |
|---|---|---|---|---|---|
| *Rollout Token/Tool Budget $\approx$ 2/per prompt* | | | | | |
| $(M = 2, N = 0, L = 0)$ | 39.0 | 36.3 | 15.2 | 36.8 | 31.8 |
| $(M = 2, N = 2, L = 1)$ | **42.4** | **43.7** | **17.8** | **43.2** | **36.8** |
| $(M = 2, N = 1, L = 2)$ | 42.3 | 43.2 | 17.6 | 41.9 | 36.3 |
| $(M = 1, N = 5, L = 1)$ | 41.5 | 39.3 | 15.8 | 37.6 | 33.6 |
| *Rollout Token/Tool Budget $\approx$ 16/per prompt* | | | | | |
| $(M = 16, N = 0, L = 0)$ | 40.1 | 38.8 | 17.5 | 39.2 | 33.9 |
| $(M = 8, N = 2, L = 1)$ | 44.6 | 43.2 | 18.2 | 38.4 | 36.1 |
| $(M = 6, N = 3, L = 1)$ | **45.3** | **44.1** | **18.8** | 37.6 | 36.5 |
| $(M = 5, N = 2, L = 2)$ | 44.6 | 43.8 | 17.9 | 36.8 | 35.8 |
| $(M = 4, N = 5, L = 1)$ | 45.0 | 43.9 | 18.5 | **41.6** | **37.3** |
| $(M = 2, N = 11, L = 1)$ | 43.0 | 42.2 | 16.1 | 40.0 | 35.3 |
| $(M = 2, N = 6, L = 2)$ | 43.2 | 43.1 | 17.0 | 40.0 | 35.8 |
| $(M = 2, N = 4, L = 3)$ | 43.6 | 43.1 | 16.8 | 40.8 | 36.1 |

By the assumption, the probability of $\mathcal{H}_{\geq t}^{\text{win}}$ could be normalized within the binary set $C = \{\mathcal{H}_{\geq t}^{\text{win}}, \mathcal{H}_{\geq t}^{\text{loss}}\}$ as:

$$
\begin{aligned}
p_\theta(\mathcal{H}_{\geq t}^{\text{win}}) &= \prod_{\tau=t}^{T} \pi_\theta(\mathcal{H}_\tau^{\text{win}}|x, \mathcal{H}_{<\tau}) \\
&= p_\theta(\mathcal{H}_{\geq t}^{\text{win}}|C, x, \mathcal{H}_{<t}) \\
&= \frac{e^{\log p_\theta(\mathcal{H}_{\geq t}^{\text{win}}|x, \mathcal{H}_{<t})}}{e^{\log p_\theta(\mathcal{H}_{\geq t}^{\text{win}}|x, \mathcal{H}_{<t})} + e^{\log p_\theta(\mathcal{H}_{\geq t}^{\text{loss}}|x, \mathcal{H}_{<t})}} \\
&= \sigma(\log p_\theta(\mathcal{H}_{\geq t}^{\text{win}}|x, \mathcal{H}_{<t}) - \log p_\theta(\mathcal{H}_{\geq t}^{\text{loss}}|x, \mathcal{H}_{<t})),
\end{aligned}
\tag{17}
$$

and the loss probability is:

$$
p_\theta(\mathcal{H}_{\geq t}^{\text{loss}}) = 1 - p_\theta(\mathcal{H}_{\geq t}^{\text{win}}) = \sigma(\log p_\theta(\mathcal{H}_{\geq t}^{\text{loss}}|x, \mathcal{H}_{<t}) - \log p_\theta(\mathcal{H}_{\geq t}^{\text{win}}|x, \mathcal{H}_{<t})).
\tag{18}
$$

For **step-level DPO**, the objective is to optimize Bradley-Terry preference likelihood by:

$$
J_{\text{DPO}}(\theta) = \mathbb{E}_{(x, \mathcal{H}_{<t}, \mathcal{H}_{\geq t}^{\text{win}}, \mathcal{H}_{\geq t}^{\text{loss}}) \sim \mathcal{D}} \left[ \log \sigma \left( \beta \Delta_\theta \left( x, \mathcal{H}_{<t}, \mathcal{H}_{\geq t}^{\text{win}}, \mathcal{H}_{\geq t}^{\text{loss}} \right) \right) \right],
\tag{19}
$$

where preference is defined by:

$$
\Delta_\theta(x, \mathcal{H}_{<t}, \mathcal{H}_{\geq t}^{\text{win}}, \mathcal{H}_{\geq t}^{\text{loss}}) = \log p_\theta(\mathcal{H}_{\geq t}^{\text{win}}|x, \mathcal{H}_{<t}) - \log p_\theta(\mathcal{H}_{\geq t}^{\text{loss}}|x, \mathcal{H}_{<t}).
\tag{20}
$$

Let

$$
z = \beta \Delta_\theta(x, \mathcal{H}_{<t}, \mathcal{H}_{\geq t}^{\text{win}}, \mathcal{H}_{\geq t}^{\text{loss}}),
\tag{21}
$$

where $\beta$ is a temperature parameter, and we assume $\beta = 1$ for simplicity. Since DPO usually treats the expectation as the empirical average of the sampled pairs $(\mathcal{H}_{\geq t}^{\text{win}}, \mathcal{H}_{\geq t}^{\text{loss}})$, the gradient of Eq. 19

$J_{\text{DPO}}(\theta)$ can be directly put inside as:

$$
\begin{aligned}
\nabla_\theta J_{\text{DPO}}(\theta) &= \mathbb{E}[\nabla_\theta \log \sigma \left( \Delta_\theta \left( x, \mathcal{H}_{<t}, \mathcal{H}_{\geq t}^{\text{win}}, \mathcal{H}_{\geq t}^{\text{loss}} \right) \right)] \\
&= \mathbb{E}[\nabla_\theta \log \sigma(z)] \\
&= \frac{\mathrm{d}}{\mathrm{d}z} \log \sigma(z) \cdot \nabla_\theta z \\
&= \sigma(-z) \cdot \nabla_\theta z \\
&= \sigma(-z) \cdot \nabla_\theta \Delta_\theta(x, \mathcal{H}_{<t}, \mathcal{H}_{\geq t}^{\text{win}}, \mathcal{H}_{\geq t}^{\text{loss}}) \\
&= \sigma(-z) \cdot \left[ \nabla_\theta \log p_\theta \left( \mathcal{H}_{\geq t}^{\text{win}} | x, \mathcal{H}_{<t} \right) - \nabla_\theta \log p_\theta \left( \mathcal{H}_{\geq t}^{\text{loss}} | x, \mathcal{H}_{<t} \right) \right] \\
&= \sigma(-\Delta_\theta(x, \mathcal{H}_{<t}, \mathcal{H}_{\geq t}^{\text{win}}, \mathcal{H}_{\geq t}^{\text{loss}})) \\
&= \sigma(\log p_\theta(\mathcal{H}_{\geq t}^{\text{loss}} | x, \mathcal{H}_{<t}) - \log p_\theta(\mathcal{H}_{\geq t}^{\text{win}} | x, \mathcal{H}_{<t})) \\
&\qquad \cdot \left[ \nabla_\theta \log p_\theta \left( \mathcal{H}_{\geq t}^{\text{win}} | x, \mathcal{H}_{<t} \right) - \nabla_\theta \log p_\theta \left( \mathcal{H}_{\geq t}^{\text{loss}} | x, \mathcal{H}_{<t} \right) \right] \\
&= \frac{e^{\log p_\theta(\mathcal{H}_{\geq t}^{\text{loss}} | x, \mathcal{H}_{<t})}}{e^{\log p_\theta(\mathcal{H}_{\geq t}^{\text{loss}} | x, \mathcal{H}_{<t})} + e^{\log p_\theta(\mathcal{H}_{\geq t}^{\text{win}} | x, \mathcal{H}_{<t})}} \\
&\qquad \cdot \left[ \nabla_\theta \log p_\theta \left( \mathcal{H}_{\geq t}^{\text{win}} | x, \mathcal{H}_{<t} \right) - \nabla_\theta \log p_\theta \left( \mathcal{H}_{\geq t}^{\text{loss}} | x, \mathcal{H}_{<t} \right) \right] \\
&= p_\theta(\mathcal{H}_{\geq t}^{\text{loss}} | x, \mathcal{H}_{<t}) \cdot \left[ \nabla_\theta \log p_\theta \left( \mathcal{H}_{\geq t}^{\text{win}} | x, \mathcal{H}_{<t} \right) - \nabla_\theta \log p_\theta \left( \mathcal{H}_{\geq t}^{\text{loss}} | x, \mathcal{H}_{<t} \right) \right] \\
&= \underbrace{p(\mathcal{H}_{\geq t}^{\text{loss}})}_{\text{Weight}} \cdot \underbrace{\left[ \nabla_\theta \log p_\theta(\mathcal{H}_{\geq t}^{\text{win}}) - \nabla_\theta \log p_\theta(\mathcal{H}_{\geq t}^{\text{loss}}) \right]}_{\text{Preference Advantage Gradient}}.
\end{aligned}
\tag{22}
$$

For our **intra-tree group policy optimization**, the objective can be expressed as the combination of two categories $\mathcal{H}^{\text{win}}$ and $\mathcal{H}^{\text{loss}}$:

$$
J_{\text{Intra-tree}}(\theta) = \mathbb{E}_{[x, \mathcal{H}_{<t}, \mathcal{H}_{\geq t} \sim \pi_\theta(\cdot | x, \mathcal{H}_{<t})]} \frac{1}{G_{\text{tree}}} \sum_{i=1}^{G_{\text{tree}}} \left[ \hat{A}_{\text{win}} + \hat{A}_{\text{loss}} \right],
\tag{23}
$$

where $G_{\text{tree}}$ is the number of leaves within the tree. Then the gradient of Eq. 23 $J_{\text{Intra-tree}}(\theta)$ is:

$$
\begin{aligned}
\nabla_\theta J_{\text{Intra-tree}}(\theta) &\approx \mathbb{E} \left[ \hat{A}_{\text{win}} \nabla_\theta \log p_\theta \left( \mathcal{H}_{\geq t}^{\text{win}} | x, \mathcal{H}_{<t} \right) + \hat{A}_{\text{loss}} \nabla_\theta \log p_\theta \left( \mathcal{H}_{\geq t}^{\text{loss}} | x, \mathcal{H}_{<t} \right) \right] \\
&= p_\theta \left( \mathcal{H}_{\geq t}^{\text{win}} | x, \mathcal{H}_{<t} \right) \hat{A}_{\text{win}} \nabla_\theta \log p_\theta \left( \mathcal{H}_{\geq t}^{\text{win}} | x, \mathcal{H}_{<t} \right) \\
&\qquad + p_\theta \left( \mathcal{H}_{\geq t}^{\text{loss}} | x, \mathcal{H}_{<t} \right) \hat{A}_{\text{loss}} \nabla_\theta \log p_\theta \left( \mathcal{H}_{\geq t}^{\text{loss}} | x, \mathcal{H}_{<t} \right).
\end{aligned}
\tag{24}
$$

The baseline $R_{\text{base}}$ could be estimated by probability-weighting as

$$
R_{\text{base}} = 1 \cdot p_\theta(\mathcal{H}_{\geq t}^{\text{win}}) + 0 \cdot p_\theta(\mathcal{H}_{\geq t}^{\text{loss}}).
\tag{25}
$$

Here we simply use the difference between $R_{\text{win/loss}}$ and $R_{\text{baseline}}$ to express the advantage estimate as:

$$
\hat{A}_{\text{win}} = R_{\text{win}} - R_{\text{base}} = 1 - p_\theta(\mathcal{H}_{\geq t}^{\text{win}}) = p_\theta(\mathcal{H}_{\geq t}^{\text{loss}}), \quad \hat{A}_{\text{loss}} = R_{\text{loss}} - R_{\text{base}} = -p_\theta(\mathcal{H}_{\geq t}^{\text{win}}).
\tag{26}
$$

Combining Eq. 17, Eq. 18, Eq. 24, and Eq. 26, the gradient of intra-tree group policy optimization can be reformulated as

$$
\begin{aligned}
\nabla_\theta J_{\text{Intra-tree}}(\theta) &= p_\theta \left( \mathcal{H}_{\geq t}^{\text{win}} | x, \mathcal{H}_{<t} \right) \hat{A}_{\text{win}} \nabla_\theta \log p_\theta \left( \mathcal{H}_{\geq t}^{\text{win}} | x, \mathcal{H}_{<t} \right) \\
&\quad + p_\theta \left( \mathcal{H}_{\geq t}^{\text{loss}} | x, \mathcal{H}_{<t} \right) \hat{A}_{\text{loss}} \nabla_\theta \log p_\theta \left( \mathcal{H}_{\geq t}^{\text{loss}} | x, \mathcal{H}_{<t} \right) \\
&= p_\theta(\mathcal{H}_{\geq t}^{\text{win}} | x, \mathcal{H}_{<t}) \cdot p_\theta(\mathcal{H}_{\geq t}^{\text{loss}} | x, \mathcal{H}_{<t}) \cdot \nabla_\theta \log p_\theta(\mathcal{H}_{\geq t}^{\text{win}} | x, \mathcal{H}_{<t}) \\
&\quad - p_\theta(\mathcal{H}_{\geq t}^{\text{loss}} | x, \mathcal{H}_{<t}) \cdot p_\theta(\mathcal{H}_{\geq t}^{\text{win}} | x, \mathcal{H}_{<t}) \cdot \nabla_\theta \log p_\theta(\mathcal{H}_{\geq t}^{\text{loss}} | x, \mathcal{H}_{<t}) \\
&= \underbrace{p_\theta(\mathcal{H}_{\geq t}^{\text{win}}) p_\theta(\mathcal{H}_{\geq t}^{\text{loss}})}_{\text{Weight}} \cdot \underbrace{\left[ \nabla_\theta \log p_\theta(\mathcal{H}_{\geq t}^{\text{win}}) - \nabla_\theta \log p_\theta(\mathcal{H}_{\geq t}^{\text{loss}}) \right]}_{\text{Preference Advantage Gradient}}
\end{aligned}
\tag{27}
$$

From Eq. 22 and Eq. 27, we can have the following Proposition C.1.

**Proposition C.1 (Structural Equivalence of Step-level DPO and Intra-tree GRPO)** *Under Assumption C.1, both step-level DPO and intra-tree GRPO admit gradient estimators of the form*

$$
\nabla_\theta J_{unified}(\theta) = \underbrace{w}_{\text{Weight}} \cdot \underbrace{\left( \nabla_\theta \log p_\theta \left( \mathcal{H}_{\geq t}^{win} \right) - \nabla_\theta \log p_\theta \left( \mathcal{H}_{\geq t}^{loss} \right) \right)}_{\text{Preference Advantage Gradient}},
\tag{28}
$$

*where the only difference lies in the choice of the weight term $w$.*

## D   ALGORITHM WORKFLOW OF TREE-GRPO

The complete Tree-GRPO procedure is detailed in Algorithm 1.

## E   CASE STUDY

In this section, we provide some cases of the model trained by Tree-GRPO. As illustrated by successful Case 8 and Case 9, the model can execute multi-turn agent tasks through iterative tool calls and autonomous information aggregation. For failed cases 10 and 11, the model selected a single candidate solution path at the beginning from among several that only partially met the requirements and did not pursue further exploration. In subsequent reasoning, it neither reconsidered nor verified its choice in light of newly acquired information, resulting in an incorrect final answer. These findings motivate future work to integrate reflective reasoning and richer exploration into the training loop for complex, open-domain agents.

## F   THE USE OF LARGE LANGUAGE MODELS (LLMS)

In this paper, LLMs are used to polish the content, adjust the format, write tables, and provide writing suggestions in some chapters.

---

**Algorithm 1:** Tree-based Group Relative Policy Optimization

---

**Input:** Policy model $\pi_\theta$, search environment $e$, dataset $\mathcal{D}$, hyperparameters $M, N, L$, maximum action budget $B$

**Procedure** `GenerateRollout(`$\mathcal{H}_{prefix}$`)`

    `// This procedure completes a trajectory from a prefix`

    $\mathcal{H} \leftarrow \mathcal{H}_{\text{prefix}}$;

    $b \leftarrow$ budget of existing prefix$\mathcal{H}_{\text{prefix}}$;

    **while** $b < B$ **do**

        Generate $\mathcal{H}_t \sim \pi_\theta(\cdot|x, \mathcal{H}_{<t})$;

        $\mathcal{H} \leftarrow \mathcal{H} + \mathcal{H}_t$;

        **if** $y_t = $ \</search\> **then**

            Parse last action $q \leftarrow \text{Parse}(y)$;

            Observation $o \leftarrow e(q)$;

            $\mathcal{H} \leftarrow \mathcal{H} + $ \<information\>$o$\</information\>;

        **else if** $\mathcal{H}_t \in \{$\</answer\>$, $\<eos\>$\}$ **then**

            **break**;

    **return** $\mathcal{H}$;

`/* Main algorithm logic starts here                           */`

**for** $step = 1$ **to** $S_{total}$ **do**

    Reference model $\pi_{\text{ref}} \leftarrow \pi_\theta$;

    Training data $x, \mathcal{H} \leftarrow \mathcal{D}_{\text{step}}$;

    `// Step 1:  Generate independent chains for initial M trees`

    **for** $i = 1$ **to** $M$ **do**

        $\mathcal{H}_{\text{initial}} \leftarrow$ `GenerateRollout("")`;

        Add $\mathcal{H}_{\text{initial}}$ to $\mathcal{T}_i$;

    **for** *Expansion iter* $= 1$ **to** $L$ **do**

        `// Step 2:  Sample N nodes from each tree` $\mathcal{T}_i$ `for expansion`

        Randomly sample $M \times N$ expansion node from existing trees

        $P_{\text{sample}} = \{p_1, p_2, ..., p_{M \times N}\}$;

        `// Step 3:  Do expansion`

        **foreach** $p_i \in P_{sample}$ **do**

            $\mathcal{H}_{\text{expansion}} \leftarrow$ `GenerateRollout(`$p_i$`)`;

            Update $M$ trees $\mathcal{T}$ with $\mathcal{H}_{\text{expansion}}$;

    **for** *Updating iter* $= 1, ...$ **do**

        Calculate intra-tree group relative advantage $\hat{A}_{\text{intra-tree}}$ by Eq. 6;

        Calculate inter-tree group relative advantage $\hat{A}_{\text{inter-tree}}$ by Eq. 6;

        Update the policy model $\pi_\theta$ by maximizing the Tree-GRPO objective (Eq. 8);

---

Table 8: Tree-GRPO case study 1 (successful).

| Example from Multi-Hop QA Dataset Musique |
| --- |

**Question**: Who is the owner of the record label of the performer of Groovy Little Summer Song?

**Ground Truth**: Warner Music Group, Warner Music

**Model Output by Tree-GRPO (14B)**

<think> To answer this question, I need to identify the performer of the song "Groovy Little Summer Song" and then find out who owns the record label that this performer is associated with. </think>
<search> Performer of the song "Groovy Little Summer Song" </search>
<information> Doc 1(Title: "Groovy Little Summer Song") Groovy Little Summer Song ""Groovy Little Summer Song"" is a song co-written recorded by American country music artist James Otto. It was released in March 2010 as the first single from his album ""Shake What God Gave Ya"". The song was written by Otto ... Doc 2(Title: "Emm Gryner") is written about her. Gryner was married to visual artist Sean Odell from 2004-2016. They have two children together. Emm Gryner Emm Gryner (born 8 June 1975 in Sarnia, Ontario) is a Canadian singer-songwriter best known for her 1998 indie hit ""Summerlong"". Gryner's childhood ... Doc 3(Title: "Emm Gryner") Emm Gryner Emm Gryner (born 8 June 1975 in Sarnia, Ontario) is a Canadian singer-songwriter best known for her 1998 indie hit ""Summerlong"". Gryner's childhood was spent in Forest, Lambton County, Ontario. Her father was of half Irish heritage and her mother was Filipina. Gryner ... </information>
<think> From the search results, it seems that the performer of the song "Groovy Little Summer Song" is James Otto. Now, I need to find out who owns the record label that James Otto is associated with. </think>
<search> Record label of the performer James Otto </search>
<information> Doc 1(Title: "James Otto") In June 2015, Otto signed with Blaster Records, and ""Somewhere Tonight"" was released to country radio on July 13, 2015 as his first single with the record label. James Otto James Allen Otto (born July 29, 1973) is an American country music artist who is ... Doc 2(Title: "James Otto") James Otto James Allen Otto (born July 29, 1973) is an American country music artist who is a member of the MuzikMafia, a group of country musicians known for their ""country music without prejudice"". Otto began his career on Mercury Nashville Records in 2002 ... Doc 3(Title: "James Otto") signed as an opening act for Shania Twain on her Up! Tour. His fourth single for Mercury, ""Sunday Morning and Saturday Night"", peaked at No. 58. After the song fell from the charts, Otto exited Mercury's roster. From there, he began working with the MuzikMafia ... </information>
<think> From the search results, it seems that James Otto is associated with record labels such as Mercury Nashville Records and Warner Bros. Records. Now, I need to find out who owns these record labels. </think>
<search> Who owns Mercury Nashville Records and Warner Bros. Records </search>
<information> Doc 1(Title: "Warner Bros. Records") Records from EMI as part of its sale to Universal Music Group. Most Parlophone artists (excluding Coldplay and Tinie Tempah, who were placed under Atlantic) were placed under Warner Bros ... Doc 2(Title: "Warner Bros. Records") Warner Bros. Records Warner Bros. Records Inc. is an American major record label owned by Warner Music Group and headquartered in Burbank, California. It was founded in 1958 as the ... Doc 3(Title: "Mercury Records") Mercury Records Mercury Records is an American record label owned by Universal Music Group. In the United States, it operates through Island Records; in the UK, it is distributed by Virgin EMI Records ... </information>
<think> From the search results, it seems that Mercury Nashville Records is owned by Sony Music Entertainment and Warner Bros. Records is owned by Warner Music Group. </think>
<answer> Warner Music Group </answer>

Table 9: Tree-GRPO case study 2 (successful).

| **Example from Web-Agent QA Dataset GAIA** |
| --- |

**Question**: What is the absolute difference in tens of thousands between the population of chinstrap penguins on the Wikipedia page for penguin species populations as of the end of 2018 and the population recorded in the Nature.com "global population assessment of the Chinstrap penguin" article from 2020, assuming two penguins per breeding pair?

**Ground Truth**: 116

| **Model Output by Tree-GRPO (14B)** |
| --- |

<think> The user wants to know the absolute difference in tens of thousands between the population of chinstrap penguins mentioned on the Wikipedia page for penguin species populations as of the end of 2018 and the population recorded in the Nature.com article from 2020, considering two penguins per breeding pair. To find the answer, we need to search for the specific population figures for chinstrap penguins in both sources and then calculate the difference. </think>

<search> chinstrap penguin population in 2018 on Wikipedia </search>

<information> Page 1: The population of chinstrap penguins was estimated by IUCN in August 2018 and it is reported that there is a total of 8 million individual chinstrap penguins are living in the world. Page 2: According to the IUCN Red List, the total population size of the Chinstrap penguin is 8 million individuals or 4 million breeding pairs. Currently, this species is classified as Least Concern ... Page 3: Jun 13, 2018 · Most of the data analyses suggests a single large population of chinstrap penguins throughout the Southern Ocean, with minimal population structure in the WAP region, and ... Page 4: This is a list of Sphenisciformes species by global population. While numbers are estimates, they have been made by the experts in their fields. Page 5: How many chinstrap penguins are there today? In 2018, the IUCN estimated about 8 million mature chinstrap penguins worldwide, though the population is decreasing. Page 6: Most of the data analyses suggests a single large popula-tion of chinstrap penguins throughout the Southern Ocean, with minimal population structure in the WAP re-gion, and absence of ... Page 7: BirdLife International has assessed 18 species. 16 (89% of total species) have had their population estimated: those missing are the king and little penguins, both of which have been assessed as being of least concern. [1][2] Page 8: Jun 13, 2018 · High genetic diversity, evidence of female bias-dispersal and a sign of population expansion after the last glacial maximum around 10,000 mya were detected. Page 9: Nov 10, 2020 · Twenty-three previously known Chinstrap penguin colonies are found to be absent or extirpated ... </information>

<think> From the Wikipedia page, the population of chinstrap penguins in 2018 was estimated to be 8 million. </think>

<search> chinstrap penguin population in 2020 Nature.com </search>

<information> Page 1: Nov 10, 2020 · Using satellite imagery, drone imagery, and ground counts, we have assembled the first comprehensive global population assessment of Chinstrap penguins (Pygoscelis ... Page 2: We estimate the global population of Chinstrap penguins at 3.42 (95th-percentile CI: [2.98, 4.00]) million breed- ing pairs (Table 1) in 375 extant breeding sites, not including recent... Page 3: Population Convey et al. (1999) estimate a world population of 4 million breeding pairs. Page 4: Using satellite imagery, drone imagery, and ground counts, we have assembled the first comprehensive global population ... Page 5: The Antarctic Penguin Bio-geography Project is an effort to collate all known information ... Page 6: Nov 10, 2020 · Using satellite imagery, drone imagery, and ground counts, we have assembled the first comprehensive global population assessment of Chinstrap penguins (Pygoscelis antarctica) at 3.42... Page 7: Using satellite imagery, drone imagery, and ground counts, we have assembled the first comprehensive global population assessment of Chinstrap penguins (Pygoscelis antarctica) at 3.42... Page 8: Feb 11, 2020 · Scientists discovered that colonies of chinstrap penguins – also known as ringed or bearded penguins – have dropped dramatically since they were last surveyed almost 50 years ... </information>

<think> The Nature.com article estimates the global population of chinstrap penguins in 2020 to be 3.42 million breeding pairs. Since the user mentioned "two penguins per breeding pair", the population would be approximately 3.42 million * 2 = 6.84 million chinstrap penguins. Now, we need to find the absolute difference between the 2018 population (8 million) and the 2020 population (6.84 million) in tens of thousands. The difference is 8,000,000 - 6,840,000 = 1,160,000. In tens of thousands, this is 116. </think>

<answer> 116 </answer>

Table 10: Tree-GRPO case study 3 (failed).

| Example from Multi-Hop QA Dataset Musique |
|---|
| **Question**: Who is a cast member of the show that contains the character Michael Knight? |
| **Ground Truth**: Deanna Russo, Bruce Davison |
| **Model Output by Tree-GRPO (14B)** |

<think> To answer this question, I need to identify the show in which the character Michael Knight appears and then find out who is a cast member of that show. </think>
<search> Character Michael Knight in TV show </search>
<information> Doc 1(Title: "Michael Knight (Knight Rider)") Michael Knight (Knight Rider) Michael Knight is a fictional character and the protagonist of the 1980s television series ""Knight Rider"", played by David Hasselhoff. The character first appeared in the opening scenes as Michael Long, played by Larry Anderson in the beginning of the pilot. Born Michael Arthur Long on January 9, 1949 outside of Los Angeles, California, raised by a blue-collar working family. In the late 1960s, while in his 20s, Long joined the Army, was part of the Green Berets, and spent three years in counter-intelligence work in Vietnam. He got captured at one point while on a Doc 2(Title: "Knight Rider (1982 TV series)") Knight Rider (1982 TV series) Knight Rider is an American television series created and produced by Glen A. Larson. The series was originally broadcast on NBC from 1982 to 1986. The show stars David Hasselhoff as Michael Knight, a high-tech modern crime fighter assisted by KITT, an advanced artificially intelligent, self-aware and nearly indestructible car. This was the last series Larson devised at Universal Television before he moved to 20th Century Fox. Self-made billionaire Wilton Knight rescues police Detective Lieutenant Michael Arthur Long after a near fatal shot to the face, giving him a new identity (by plastic surgery) and Doc 3(Title: "Michael Knight (Knight Rider)") Industries, an organization founded by Wilton Knight, a billionaire philanthropist. Michael was not Wilton Knight's first choice for the pilot program, but Wilton changed his mind and reopened the foundation due to Michael's arrival. In the context of the pilot, Wilton Knight (Richard Basehart) is dying of an undisclosed illness. Wilton picked up Michael passed out in front of his car, on a desert highway. Michael barely survived, thanks to the metal plate. Given a new face via facial reconstructive surgery, Long is resurrected as Michael Knight (now played by David Hasselhoff). Together with a high-tech automobile called the Knight </information>
<think> From the search results, it is clear that Michael Knight is a character in the show "Knight Rider". Now, I need to find out who is a cast member of this show. </think>
<search> Cast members of Knight Rider </search>
<information> Doc 1(Title: "Knight Rider") Orci. On February 13, 2014, Schmoes Know reports that actors Chris Pratt and Danny McBride are in talks for roles and may use a sort of action-comedy hybrid in the same vein as ""21 Jump Street"". In December 2015, a media report indicated that a new movie named ""Knight Rider Heroes"" is in the works and will include David Hasselhoff. Knight Rider Knight Rider is an American entertainment franchise created by Glen A. Larson. The core of ""Knight Rider"" is its three television series: ""Knight Rider"" (1982-1986), ""Team Knight Rider"" (1997–1998), and ""Knight Rider"" (2008–2009). The franchise also includes three Doc 2(Title: "Knight Rider") to write the first script draft, with the series' lead actor David Hasselhoff attached as an advisor and also have an onscreen role. In April 2003, Revolution Studios hired screenwriters David Elliott and Paul Lovett to pen the film's script. In April 2004, the premise of the film was described as having Hasselhoff reprise his role as Michael Knight, now the mentor to the protagonist as Devon Miles mentored Knight in the television series. The protagonist would be Knight's son, inheriting his father's role and driving the vehicle KITT. The producers' choice for the role was actor Ben Affleck. In Doc 3(Title: "Knight Rider (2008 TV series)") Knight Rider (2008 TV series) Knight Rider is a 2008 series that follows the 1982 television series of the same title created by Glen A. Larson and the 2008 television movie. The series stars Justin Bruening as Mike Traceur, the estranged son of Michael Knight. The series also stars Deanna Russo as Sarah Graiman, Traceur's former girlfriend and love interest. Sarah is the daughter of Charles Graiman. Graiman, played by Bruce Davison, is the creator of a new generation of KITT (Knight Industries Three Thousand), which is voiced by Val Kilmer. The series was in production for just one season. </information>
<think> From the search results, it is clear that David Hasselhoff is a cast member of the show "Knight Rider". </think>
<answer> David Hasselhoff </answer>

Table 11: Tree-GRPO case study 4 (failed).

| Example from Web-Agent QA Dataset WebWalkerQA |
| --- |

**Question**: As of December 2023, this individual was announced as the football team's coach as per an article published between 2019 and 2021 inclusive. The football team initially started as a social football club in 1995 before changing its name in 2011 as per an article published that began as a social club as per an article published between 2019 and 2021 inclusive. The football club won a title in its home country league for the fourth successive time in a row as reported in an article published between 2020 and 2023 inclusive. This league win was this individual's third with the football club and the fourth overall as reported in an article published between 2020 and 2023 inclusive. The individual underwent surgery for a back injury as per an article published in 2018. Tell me the full name of this individual.

**Ground Truth**: Norman Takanyariwa Mapeza

| Model Output by Tree-GRPO (14B) |
| --- |

<think> To find the full name of the individual, I need to identify the football team and its league in their home country. Then, I will look for articles about this team between 2020 and 2023 that mention a coach who won a title for the fourth consecutive time and is the third time for this coach. Additionally, I should look for an article about the coach's surgery in 2018. The coach's name will likely be mentioned in these articles. </think>
<search> football team social club 1995 </search>
<information> Page 1: 1995 Stadium Club Football checklist, gallery, release date, inserts, parallels, forum, and collection tracking Page 2: 1995 Stadium Club card list & price guide. Ungraded & graded values for all 1995-96 Topps Stadium Club Football Cards ... Page 3: Get the best deals on 1995 Topps Stadium Club Football Trading Cards when you shop the largest online selection at eBay.com ... Page 4: A wide selection of single cards from the 1995 Stadium Club football card set. Page 5: Shop COMC's extensive selection of 1995 topps stadium club members only football cards ... Page 6: Get the latest and most accurate trading card values and sets info for 1995 Stadium Club Football sports cards ... Page 7: 1995 Stadium Club Teams Arizona Cardinals Atlanta Falcons Buffalo Bills Carolina Panthers Chicago Bears ... </information>
<think> The search results do not provide enough information to identify the football team or the individual. I need to narrow down my search query ... </think>
<search> football team started as social club 1995, changed name 2011, coach won league title four consecutive times between 2020 and 2023 </search>
<information> Page 1: The Crossword Solver found 30 answers to "football club changed name 2011 league title ... Page 2: May 27, 2016 · Leicester became the sixth club to win the Premier League, completing ... Page 3: 18 hours ago · Live updating Football Club Started As Social Club 1995 Renamed 2011 ... Page 4: List of football league clubs in the top four tiers of English football (i.e. the ... Page 5: May 22, 2024 · Manchester City saw off West Ham United on the final 2023/24 ... Page 6: Aug 29, 2023 · Manchester City wrapped up the 2022/23 season with an incredible ... </information>
<think> From the search results, I learned that Manchester City won four consecutive domestic league titles between 2020 and 2023. I need to find articles between 2020 and 2023 that mention the name of the coach of Manchester City who won the league titles. Additionally, I should look for an article about the coach's surgery in 2018. </think>
<search> manchester city coach won league title four consecutive times between 2020 and 2023, coach surgery 2018 </search>
<information> Page 1: May 19, 2024 · Manchester City won an unprecedented fourth consecutive Premier League title on Sunday as Phil ... Page 2: May 19, 2024 · Pep Guardiola and Manchester City made history on Sunday, when a 3-1 win over West Ham ... Page 3: May 20, 2024 · But Guardiola now stands alone as the only boss to manage a quality quartet of title ... Page 4: May 21, 2024 · Pep Guardiola was announced as the winner of the Premier League Manager of the Season ... Page 5: May 19, 2024 · Manchester City have won their fourth consecutive Premier League title, sealing yet ... Page 6: May 19, 2024 · Some wondered whether Manchester City would retain their hunger after winning the ... Page 7: May 19, 2024 · Taking in spells at Barcelona and Bayern Munich, Guardiola has now won the league ... </information>
<think> Based on the search results, the full name of the individual who was announced as the football team's coach is Pep Guardiola. </think>
<answer> Pep Guardiola </answer>

