# OpenReview forum: "Tree Search for LLM Agent Reinforcement Learning"
_ICLR.cc/2026/Conference — ICLR 2026 Poster_

### Official Review · Reviewer_uD4w · 2025-10-30

**Soundness:** 3
**Presentation:** 3
**Contribution:** 3
**Rating:** 6
**Confidence:** 3

**Summary:**

This paper introduces Tree-GRPO, a novel method designed to enhance LLM agents in long-term, multi-turn RL tasks. The authors address two critical challenges in current agentic RL: the heavy computational budget associated with LLM rollouts, and the sparse supervision problem arising from outcome-only rewards.

1. Tree-GRPO proposes a tree-search based online rollout strategy where each tree node represents a complete agent interaction step (Thought-Action-Observation). By sharing common prefixes within the tree structure, the method significantly increases the number of effective rollouts under a fixed token/tool-call budget.
2. This method leverages the tree structure to naturally derive fine-grained, step-wise process supervision signals purely from outcome rewards by estimating grouped relative advantages at both intra-tree and inter-tree levels.
3. The paper theoretically demonstrates that intra-tree level group relative policy optimization is equivalent to step-level direct preference learning.
4. Empirical evaluations across 11 datasets spanning Single-Hop, Multi-Hop, and hard Web-Agent QA tasks, using various LLM models, demonstrate Tree-GRPO's superiority over chain-based RL methods. Notably, it achieves higher performance with a significantly reduced rollout budget, particularly for smaller models and in complex multi-hop interactions.
5. The paper also provides ablation studies on tree node granularity and tree structure parameters, as well as analyses on budget efficiency and the method's ability to encourage longer, more complex interactions.

**Strengths:**

1. The paper presents a relatively original approach by integrating tree-search sampling with GRPO for online LLM agent RL. The key innovation is the agent step-level node definition within the tree, which is a clever adaptation for efficient and meaningful process supervision in multi-turn agent tasks.
2. The overall quality of the paper is exceptional. The method is clearly described, and the theoretical analysis provides valuable insights into why the method works. The experimental evaluation is thorough, covering different model families (Qwen, Llama), multiple sizes (1.5B-14B), and diverse task types (Single-Hop QA, Multi-Hop QA and Web-Agent QA). The detailed ablation studies, especially on the granularity of tree nodes (token/sentence vs. agent step) and tree structure parameters (M, N, L), examine important design choices.
3. The paper is written with outstanding clarity. The explanations are precise, the technical details are well-articulated, and the figures effectively convey complex ideas.
4. The work offers substantial significance to the field. It provides a highly effective and efficient method for training LLM agents, addressing the critical issues of sparse supervision and high computational costs. By showing how to derive fine-grained process signals from outcome rewards in an online setting, it opens new avenues for scalable and robust agent learning. The ability to encourage longer, more complex interactions is particularly important for the development of next-generation foundation models.

**Weaknesses:**

1. Detailed Breakdown of Rollout Budget Savings: Although the paper emphasizes "less rollout budget" and shows its effectiveness, a direct quantitative comparison of the actual training time and memory consumption (e.g., GPU hours, peak memory usage) between Tree-GRPO and chain-based GRPO would provide a more complete picture of its efficiency. While "less rollout budget" implies efficiency, explicit benchmarks on wall-clock time for training on large models would solidify this claim, especially since tree search itself can introduce overheads. This would complement the token/tool-call budget analysis.
2. Restrictive Theoretical Assumptions: Assumption 3.1 (binary preference setting) is quite strong and unlikely to hold with continuous outcome rewards. The theoretical analysis would be more convincing with relaxed assumptions or empirical validation that the approximation holds reasonably well.
3. Inconsistent Improvements: Performance gains vary considerably across settings. On single-hop QA with larger models (14B), improvements are marginal (~1%). On web-agent tasks, gains are limited by training data quality. This suggests the method's benefits are context-dependent.
4. Experimental Limitations: There lackscomparison with explicit process reward models, which would be a natural baseline. There is also missing analysis of computational overhead for tree construction/management during training.
5. Sensitivity Analysis for Key Hyperparameters Beyond Tree Structure: The paper conducts an excellent ablation on tree structure parameters (M, N, L) and shows LR warmup sensitivity. However, a more comprehensive sensitivity analysis for other critical hyperparameters, such as the KL coefficient or max response length across different tasks, could provide further guidance for practitioners on how to tune Tree-GRPO effectively. For example, how robust is the method to changes in the KL regularization term, which is often crucial in RL?

**Questions:**

1. In Figure 1, the paper claims "less rollout budget (both on tokens and tool-calls)", and mentions achieving 1.5x samples under the same budget later. Does there exist a more detailed breakdown of how these savings in tokens and tool-calls are realized in practice?
2. What is the actual wall-clock time comparison between chain-based and tree-based training? Considering the complexity of tree management, does this offset the token savings?
3. How does your implicit process supervision compare to explicit process reward models when both are available?

---

> ### Author Response · Authors · 2025-11-21
> **Official Comment by Authors (1/2)**
>
> > **W1&Q1: More details of how savings in tokens and tool-calls are realized.**
>
> Thanks for the question and suggestion. As described in `Section 3.1`, `Eq. (4)` of the paper, the expected rollout budget of the tree search is controlled by `M,N,L`:
> $$
> \mathbb{E}[B_\text{tree}] = M \cdot B + L \cdot N \cdot B / 2.
> $$
> In the practical implementation, by constraining the maximum tree depth (*i.e.*, the agent steps), our tree-based sampling method (executed by `1.init`, `2.sample`, `3.expansion`) can achieve like 1.5× samples with a “less rollout budget” **in statistical terms**.
>
> Here we provide more detailed statistical data in the table below, where the model is Qwen2.5-3b trained on multi-hop QA. Under a comparable rollout budget (*e.g.*, `RolloutNum=4` in chain-based sampling), the tree-based configuration (`M = 2, N = 2, L = 1`) is capable of sampling 6 rollouts.  For chain-based sampling with the same `RolloutNum=6`, it would require nearly 1.5x rollout budget.
>
> | **Setting (Qwen2.5-3b on Multi-hop QA)** |  **Tool-calls/Prompt**  |    **Tokens/Prompt**     |
> | ---------------------------------------- | :---------------------: | :----------------------: |
> | Chain-based ($RolloutNum=4,M=4$)         |           7.4           |           4448           |
> | Chain-based ($RolloutNum=6,M=6$)         |          11.3           |           6688           |
> | Tree-based ($RolloutNum=6,M=2,N=2,L=1$)  | 7.5 ($\downarrow 34\\%$) | 4032 ($\downarrow 40\\%$) |
> | Chain-based ($RolloutNum=8,M=8$)         |          15.0           |           8160           |
> | Tree-based ($RolloutNum=8,M=2,N=3,L=1$)  | 9.4 ($\downarrow 37\\%$) | 5040 ($\downarrow 38\\%$) |
>
> ---
>
> > **W1&Q2: What is the actual wall-clock time comparison between chain-based and tree-based method?**
>
> Thanks for pointing this out. In fact, current LLM inference engines (*e.g.*, vLLM, SGLang) do not support flexible, customized tree search, especially the agent step-level tree search proposed in this paper. Our implementation is still built upon the standard inference interfaces, where tree-structured node sampling and expansion are achieved by dynamically maintaining the prefix inputs.
>
> This means that, although our method can save approximately 1.5× rollout budget, it still requires at least two calls of inference engine interface (`L=1`). Consequently, although a **substantial reduction in tool calls and tokens is achieved, the actual wall‑clock time does not improve in proportion to these savings**. We conduct our comparative experiments using the vLLM v1 engine on Nvidia H20 GPUs.  As shown in the table, the improvement of the tree-based method in terms of average consuming time per rollout is relatively small (`20.1 s/traj`. vs. `21.4 s/traj.`). However, compared with the chain-based method with `RolloutNum=16` which has a lower evaluation metric (`36.8` vs. `33.9`), the tree-based method still **significantly saves rollout time consumption** for one RL step (`120.7 s` vs. `300.6 s`). Enabling existing inference engines to support tree-structured sampling should be a valuable research and engineering direction for further work.
>
> | **Setting (Qwen2.5-3b on Multi-hop QA)** | **Rollout Time/Train Step (s)** | **Rollout Time/Traj. (s)** | **Avg. Test Score** |
> | ---------------------------------------- | :-----------------------------: | :------------------------: | :-----------------: |
> | Chain-based ($RolloutNum=4,M=4$)         |              85.4               |            21.4            |        31.8         |
> | Chain-based ($RolloutNum=6,M=6$)         |              129.9              |            21.6            |        31.6         |
> | Tree-based ($RolloutNum=6,M=2,N=2,L=1$)  |              120.7              |            20.1            |      **36.8**       |
> | Chain-based ($RolloutNum=16,M=16$)       |              300.6              |            18.8            |        33.9         |

---

> > ### Author Response · Authors · 2025-11-21
> > **Official Comment by Authors (2/2)**
> >
> > > **W4&Q3: Lacks comparison with explicit process reward model.**
> >
> > Current Process Reward Models (PRM) can by grouped into two categories: **(i)** the PRM trained on expensive step-level human annotations [1]; **(ii)** the implicit PRM that automatically learning from outcome rewards [2], or estimating prefix-based value function $Q^{\\pi}$ via MCTS [3] and structured sampling [4]. To the best of our knowledge, there is no category (i) explicit PRM for the agent task now due to the large amount of annotated data required. The category (ii) shares certain conceptual similarities with our approach.
> >
> > Here we provide further comparisons with the two works with explicit process supervision. `ReasonRAG` [5]  constructs step-wise, process-supervised DPO data through an automated approach. `StepSearch` [6] performs step-wise reward based policy optimization by constructing sub-question search trajectory data. The results show that our Tree-GRPO outperforms both explicit step-wise supervision methods across multi-hop QA datasets.
> >
> > | **Method (Qwen2.5-3b)** | **Hotpot** | **2Wiki** | **Musiq** | **Bamb** | **Avg.** |
> > | ----------------------- | :--------: | :-------: | :-------: | :------: | :------: |
> > | ReasonRAG [5]           |    38.4    |   43.6    |   12.8    |   36.0   |   32.7   |
> > | StepSearch [6]          |    34.5    |   32.0    |   17.4    |   34.4   |   29.6   |
> > | Tree-GRPO               |  **42.4**  | **43.7**  | **17.8**  | **43.2** | **36.8** |
> >
> > [1] Lightman H, et al. Let's Verify Step by Step
> >
> > [2] Cui G, Process Reinforcement through Implicit Rewards
> >
> > [3] Wang P, et al. Math-Shepherd: Veirfy and Reinforce LLMs Step-by-step without Human Annotations
> >
> > [4] Luo L, et al. Improve Mathematical Reasoning in Language Models by Automated Process Supervision
> >
> > [5] Wang W, et al. Process vs. Outcome Reward: Which is Better for Agentic RAG Reinforcement Learning
> >
> > [6] Wang Z, et al. StepSearch: Igniting LLMs Search Ability via Step-wise Proximal Policy Optimization
> >
> > ---
> >
> > > **W2: Restrctive theoretical assumptions of binary preference seting.**
> >
> > Our theoretical framework is specifically designed to align with the **step-level DPO** setting. In standard DPO, supervision is provided as **pairwise preferences** between two trajectories or responses $(\mathcal{H}^{win}, \mathcal{H}^{loss})$ with a binary label indicating which one is prefered. To match this regime, our Assumption 3.1 (binary preference setting) is also proposed along with a binary discrete reward setting. If changing it to continuous rewards, we need another theoretical framework to explain it.
> >
> > ---
> >
> > > **W3: Inconsistent improvements on single-hop QA and web-agent QA.**
> >
> > As discussed in `Section 4.2`, the improvement in single-hop QA is quite limited because the agent action turns are few, thus the **impact of the process signal is not substantial**. As for the web-agent QA, the model shows a relatively clear improvement on GAIA. However, its performance on benchmark like BrowseComp is indeed heavily bounded by training data limitations. We are currently working on collecting more complex training data to alleviate this issue.
> >
> > ---
> >
> > > **W5: How robust is the method to changes in KL term?**
> >
> > Thanks for the suggestion. We extend the hyperparameter ablation experiments to the KL coefficient here. In contrast to tasks such as mathematical reasoning, removing the KL term does not lead to performance improvement in our setting. An appropriately weighted KL term can yield better performance, whereas an overly strong KL constraint can limit the model’s learning capacity. Our tree-based approach maintains superior performance across various configurations for both Qwen2.5-3b and Llama3.2-3b-it, and the effect of the KL term appears to follow the same trend.
> >
> > | **Hyper-parameter Setting**           | **Chain-based** (Avg. Test Score) | **Tree-based** (Avg. Test Score) |
> > | ------------------------------------- | :-------------------------------: | :------------------------------: |
> > | Qwen2.5-3b $\beta_\text{KL} = 0$      |               30.9                |               34.3               |
> > | Qwen2.5-3b $\beta_\text{KL} = 0.001$  |               31.8                |               36.8               |
> > | Qwen2.5-3b $\beta_\text{KL} = 0.01$   |               29.8                |               32.2               |
> > | Llama3.2-3b $\beta_\text{KL} = 0$     |               23.4                |               36.8               |
> > | Llama3.2-3b $\beta_\text{KL} = 0.001$ |               26.7                |               36.8               |
> > | Llama3.2-3b $\beta_\text{KL} = 0.01$  |               23.4                |               29.7               |
> >
> > The related part is updated in `Section 4.3 Chain-based vs. Tree-based Beyond Performance` and `Figure 6` of the revised version.
> >
> > ---
> >
> > We sincerely appreciate your feedback. Hope the above response can address your concerns. We are pleased to provide further clarification if you have any further questions!

---

> ### Author Response · Authors · 2025-11-25
> **Looking Forward to Your Reply**
>
> Thank you once again for taking the time to review our paper and for providing constructive feedback.
>
> We have carefully considered each of your comments and have provided detailed responses. We sincerely hope that our efforts adequately address your concerns and contribute positively to your evaluation.
>
> As the author–reviewer discussion period progresses, we would greatly appreciate any further feedback you may have. If you have any additional questions or require any clarifications, please do not hesitate to reach out to us.

---

> > ### Comment · Reviewer_uD4w · 2025-11-26
> >
> > Thanks for the detailed responses. The authors have addressed my major concerns. I’m glad to see the ablation study has been extended to the KL coefficient.

---

> > > ### Author Response · Authors · 2025-11-26
> > > **Thanks for The Support**
> > >
> > > We truly appreciate your continued engagement during the rebuttal process, which helps us improve the manuscript.
> > > If you have any additional comments or questions, we would be happy to address them.

---

### Official Review · Reviewer_TDPe · 2025-10-31

**Soundness:** 3
**Presentation:** 3
**Contribution:** 2
**Rating:** 4
**Confidence:** 3

**Summary:**

The paper presents a new efficient tree-search rollout approach for training an RL-based LLM agent in which each node of the rollout tree represents a complete thought-action-observation step of the LLM agent. This tree-search based method is shown to obtain more samples than existing chain-based method. The paper then introduces a group-relative advantage estimation at both intra and inter-tree levels, incorporating preference objectives while leveraging rollouts from all trees, allowing for a more stable training. Experiments across different settings including multi-hop QA, single-hop QA, and web-agent QA are conducted, showing improved performance of the proposed method compared to baselines.

**Strengths:**

1. The paper is clearly written. Explanations on how rollout search trees are built and how group relative advantages are computed are easy to follow.
2. The idea of rollout search trees based on a complete thought-action-observation step is promising as it is shown to be cost-effective in collecting samples.
3. Experiments are extensive. Results on various QA settings show that the proposed method outperforms other SOTA baselines in most of the evaluated settings.

**Weaknesses:**

1. In sparse-reward RL, there is a long line of research on credit assignments or reward redistributions. There lacks discussions on why these existing sparse-reward RL algorithms were not applied in this RL-based LLM agent.
2. In the proposed tree-based credit assignment method, it appears that stepwise advantages are identical across all time steps within each trajectory. This uniform credit distribution is quite simplistic and may fail to capture the varied contributions of different steps to overall performance.

**Questions:**

1. Could you please justify why uniform distribution of credits across different steps is reasonable in this learning setting?
2. Can you provide some thoughts on applying existing sparse-reward RL methods for LLM agents?

---

> ### Author Response · Authors · 2025-11-21
> **Official Comment by Authors (1/2)**
>
> > **W2&Q1: Identical credits/advantages across all steps are simplistic and may fail to capture varied contributions. Please justify the reason.**
>
> In parrallel independent chain-based trajectories, assigning identical credits to different steps indeed fails to capture their varied contributions. However, under tree-structured sampling, while the credit allocation remains consistent along each root-to-leaf trajectory, **the advantages result in varied contributions at different steps through branches.**
>
> For ease of understanding, we present a simple example below: consider a tree structure with three nodes `A,B,C`, where the advantage of trajectory `(A,B)` is uniformly assigned a value of `1`, whereas that of `(A,C)` is assigned `-1`.
>
> ```
>         ┌─> B (1) √
> A (-1, 1)
>         └─> C (-1) x
> ```
>
> From the perspective of a single trajectory, two steps such as `(A,B)` are assigned identical credits, and thus their individual contributions cannot be distinguished. **While from the perspective of the entire tree structure**, the gradient contributions $\hat{A} \nabla_{\theta} \log p_\theta$ from the two branches to shared node `A` are added to zero. Then the final optimization objective can be regarded as a step-level preference learning between `B` and `C` branches. Based on such properties of tree-structured sampling, although the credit assignment remains at the trajectory level, the optimization objective can still **capture varied contributions across steps in different branches with different granularity**.
>
> In fact, we also try to directly estimate different advantages for different steps within each trajectory. We explore three approaches:
>
> - The first is the differential method similar to `TreeRL` [1], where the advantage of each node is calculated as the difference between its mean reward and the mean reward of its parent node and the root node:
> $$
> \hat{A}\_\text{TreeRL}(p) = r\_\text{mean}(p) - r\_\text{mean}(\text{father}(p)) + r\_\text{mean}(p) - r\_\text{mean}(\text{root}),
> $$
> $$
> r\_\text{mean}(p) = \frac{1}{\text{subtree}(p)}\sum_i^{\text{subtree}(p)} \text{reward}(i).
> $$
>
> - The second is similar to `TreePO` [2], where the mean reward of each node is normalized within the group formed by all of its ancestor nodes:
> $$
> \hat{A}\_\text{TreePO}(p) = \frac{\sum\_i^{\text{Ancestor}(p)} r\_\text{mean}(i)}{|\text{Ancestor}(p)|\cdot std(\lbrace r \_\text{mean}(i)\rbrace^\text{Ancestor(p)})},
> $$
> $$
> r\_\text{mean}(p) = \frac{1}{\text{subtree}(p)}\sum\_i^{\text{subtree}(p)} \text{reward}(i).
> $$
>
>
> - The third is a variant of `GSPO` [3], in which GSPO estimates importance ratio over the entire trajectory, whereas we try to estimate the importance ratio at each individual step:
> $$
> \hat{ir}\_\text{step-GSPO} = \left(\frac{\pi\_\theta(\mathcal{H\_{\ge t}|\mathcal{H}\_{<t},x})}{\pi_{\theta\_\text{old}}(\mathcal{H\_{\ge t}|\mathcal{H}\_{<t},x})}\right)^{\frac{1}{|\mathcal{H\_{\ge t}}|}}.
> $$
>
>
>
> | Credit Setting (Qwen2.5-3b)       |  Hotpot  |  2Wiki   |  Musiq   |   Bamb   |   Avg.   |
> | --------------------------------- | :------: | :------: | :------: | :------: | :------: |
> | TreeRL-like [1] (Eq.1)            | collapse | collapse | collapse | collapse | collapse |
> | TreePO-like [2] (Eq.3)            |   38.6   |   38.2   |   15.6   |   37.2   |   32.4   |
> | Step-GSPO [3] (tree-based) (Eq.5) |   35.3   |   37.1   |   11.3   |   29.6   |   28.3   |
> | Tree-GRPO (ours)                  |   42.4   |   43.7   |   17.8   |   43.2   |   36.8   |
>
> Our experiments show that the `TreeRL-like` differential method is highly susceptible to reward hacking, as the model tends to drive all differential baselines toward `0`. The `TreePO-like` method and `step-level GSPO` both perform worse than directly applying `vanilla GRPO` within the tree-structured groups. Based on the above analysis and experiments, we find that directly assigning distinct credits to different steps without step-level ground truth is both tricky and challenging. **The properties of tree-based sampling allow us to provide distinct step-level gradient signals while still retaining trajectory-level rewards**. Therefore, although credits are uniformly assigned to different steps within a single trajectory, **the aggregation over multiple trajectories with different branch structures naturally results in varied credit distribution across steps**, ensuring that step-level differences are still captured in the overall optimization.
>
> [1] Hou Z, et al. TreeRL: LLM Reinforcement Learning with On-Policy Tree Search
>
> [2] Li Yi, et al. TreePO: Bridging the Gap of Policy Optimization and Efficacy and Inference Efficiency with Heuristic Tree-based Modeling
>
> [3] Zheng C, et al. Group Sequence Policy Optimization

---

> ### Author Response · Authors · 2025-11-21
> **Official Comment by Authors (2/2)**
>
> > **W1&Q2: Provide some thoughts on existing sparse-reward RL methods for LLM agents.**
>
> Thanks for the valuable question and suggestion.  Here we first try to summarize existing RL methods designed for sparse-reward settings. Existing sparse-reward RL methods can generally be divided into three categories: **(i) Exploration**-oriented methods mitigate sparse rewards by encouraging the model to visit potentially rewarding states via curiosity [4], goal-conditioned replay [5], etc. **(ii) Reward-shaping** methods [6,7] provide additional intermediate supervision signals by decomposition or predicting process feedback. **(iii) Credit assignment** methods [8] redistribute outcome rewards to earlier steps through heuristic mechanisms.
>
> Consider applying these methods to LLM agent RL, we also discuss them in terms of these three categories. **(i) Exploration**: Due to the strong prior knowledge from base LLM model and the highly structured agent action space, the search space of an LLM agent is greatly constrained. Hence that the LLM agent starts with a policy that is far from random, and exploration is not a primary bottleneck in current LLM agent RL. **(ii) Reward-shaping**: For LLM/LLM agent RL, recent studies achieve reward shaping mainly by Process Reward Model (PRM) to generate step/token-level feedback. Among these, explicit PRMs [9] rely on step-level human annotation are not well suited for scaling to general scenarios. Some works [10] also try to jointly train the policy model together with an implicit PRM, while the implicit PRM model training still suffers from the sparse-reward problem. **(iii) Credit assignment**: In contrast to traditional RL, which samples from a high-dimensional and unstructured state–action space, LLM agents sample within a highly structured and semantically constrained action space. This makes it easier to obtain meaningful samples from heuristic structures (*e.g.,* tree), and consequently trajectory decomposition and credit assignment are considerably more effective. **(iv) Unique challenges**: In addition to the analyses of sparse-reward problems in existing works, LLM agent RL also faces unique challenges. For example, traditional RL can perform large-scale parallel sampling and Monte Carlo estimation, whereas such practices are unacceptable  for LLMs due to the high cost. These constraints or characteristics make it hard to directly applying existing sparse-reward RL methods to LLM agent. **In summary**, while existing sparse‑reward RL and LLM‑agent RL share similar problem-solving paradigm, the unique traits of LLM agents (strong prior knowledge, highly structured search space, expensive costs, etc.) require specific solutions.
>
> Here we also provide a comparison of several recent works on LLM RL that address sparse rewards. `VinePPO` [13] leverages Monte Carlo estimation to optimize `PPO` [12] and provide better credit assignment. `ReasonRAG` [14]  constructs step-wise, process-supervised DPO data through an automated approach. `StepSearch` [15] performs step-wise reward based policy optimization by constructing sub-question search trajectory data. Note that * means the impementation is is provided by us, and † means the results reported in the original publication. The results show that our Tree-GRPO performs best in our setting.
> |**Method (Qwen2.5-3b)**|**Hotpot**|**2Wiki**|**Musiq**|**Bamb**|**Avg.**|
> |---| :---: | :---: | :---: | :---: | :---: |
> |PPO† [12]  |  37.1 | 38.7 |15.0|32.3|30.8|
> |VinePPO* [13] ($K=3$)|38.2|41.9|13.8|38.6|33.1|
> |ReasonRAG† [14]|38.4|43.6|12.8|36.0|32.7|
> |StepSearch† [15]|34.5|32.0|17.4|34.4|29.6|
> |Tree-GRPO (ours)|**42.4**|**43.7**|**17.8**|**43.2**|**36.8**|
>
> The related part is updated in `Section 5 Related Work` (line 476-501) of the revised version.
>
> [4] Pathak D, et al. Curiosity-driven Exploration by Self-supervised Prediction
>
> [5] Andrychowicz M, et al. Hindsight Experience Replay
>
> [6] NG A, et al. Policy Invariance Under Reward Transformations: Theory and Application to Reward Shaping
>
> [7] Arjona-Medina J, et al. RUDDER: Return Decomposition for Delayed Rewards
>
> [8] Huang C, et al. Optimizing Agent Behavior over Long Time Scales by Transporting Value
>
> [9] Lightman H, et al. Let's Verify Step by Step
>
> [10] Cui G, et al. Process Reinforcement through Implicit Rewards
>
> [11] Wang P, et al. Math-Shepherd: Veirfy and Reinforce LLMs Step-by-step without Human Annotations
>
> [12] Schulman J, et al. Proximal Policy Optimization Algorithms
>
> [13] Kazemnejad A, et al. VinePPO: Unlocking RL Potential For LLM Reasoning Through Refined Credit Assignment
>
> [14] Wang W, et al. Process vs. Outcome Reward: Which is Better for Agentic RAG Reinforcement Learning
>
> [15] Wang Z, et al. StepSearch: Igniting LLMs Search Ability via Step-wise Proximal Policy Optimization
>
> ---
> We sincerely appreciate your feedback. Hope the above response can address your concerns. We are pleased to provide further clarification if you have any further questions!

---

> ### Author Response · Authors · 2025-11-25
> **Looking Forward to Your Reply**
>
> Thank you once again for taking the time to review our paper and for providing constructive feedback.
>
> We have carefully considered each of your comments and have provided detailed responses. We sincerely hope that our efforts adequately address your concerns and contribute positively to your evaluation.
>
> As the author–reviewer discussion period progresses, we would greatly appreciate any further feedback you may have. If you have any additional questions or require any clarifications, please do not hesitate to reach out to us.

---

> ### Author Response · Authors · 2025-11-27
> **Kindly Reminder for Your Feedback**
>
> Dear Reviewer TDPe,
>
> We appreciate your time and effort to review our work. We have carefully considered each of your comments and done our utmost to address your concerns.
>
> As the rebuttal period draws to a close, we would like to kindly remind that if you could review our responses and let us know whether it address your points, or if there is any additional clarification needed.
>
> Best,
>
> The authors of Paper 1473

---

### Official Review · Reviewer_PcpF · 2025-11-01

**Soundness:** 3
**Presentation:** 3
**Contribution:** 3
**Rating:** 6
**Confidence:** 3

**Summary:**

The paper "Tree Search for LLM Agent Reinforcement Learning" introduces Tree-based Group Relative Policy Optimization (Tree-GRPO), a novel method to address the high costs and sparse supervision challenges in training LLM agents for multi-turn tasks . Instead of inefficient "chain-based" rollouts, Tree-GRPO employs a tree-search sampling strategy where each node represents a complete agent step (Thought, Action, Observation). This structure significantly improves budget efficiency by sharing common prefixes and, crucially, allows the model to derive fine-grained, step-level process supervision signals directly from sparse outcome rewards by comparing the performance of different branches. The paper's key contributions include this highly efficient agent-step-level tree-search method, an implicit preference learning objective that is theoretically shown to be structurally equivalent to step-level DPO , and strong empirical results demonstrating superior performance over baselines while using as little as one-quarter of the rollout budget.

**Strengths:**

- The paper re-frames the inefficient "chain-based" sampling process  as a problem that can be optimized with a "tree-based" shared-prefix structure.
- The authors don't just propose a method; they provide a theoretical justification (Proposition 3.1) that connects their intra-tree advantage estimation to the well-established DPO framework , lending it significant credibility.
- The quality is further reinforced by a series of insightful ablation studies. These include testing performance under different rollout budgets, analyzing the contribution of the $\hat{A}_{intra-tree}$ and $\hat{A}_{inter-tree}$ advantage components, and comparing against the intuitive token-level tree search.

**Weaknesses:**

- While Tree-GRPO showed consistent improvements, the authors note that for highly challenging web-agent benchmarks like BrowseComp, the gains from RL (both chain-based and tree-based) were marginal. They attribute this primarily to the limited quality and difficulty of the available training data rather than a failure of the method itself .
- The method introduces new hyperparameters that govern the tree structure, specifically the number of initial trees ($M$), the number of nodes to expand ($N$), and the number of expansion iterations ($L$). This will introduce additional hyperparameter choice under different situations.

**Questions:**

- In Table 4, it seems that intra-tree advantage will lead to collapse and the main performance gain is from the inter-tree. So the mainly the advantage estimated by different initialization contribute most to the performance. So, it seems that large $M$ is more important than large $N$ and large $L$. Is that understanding correct?
- How do you choose M,N,L in prior? In practice, it seems that $L=1$ is the best choice. But how to balance $M$ and $N$ given the same computational budget? For example, to choose larger M or larger N given the same budget?

---

> ### Author Response · Authors · 2025-11-21
> **Official Comment by Authors (1/2)**
>
> > **Q1: The relation between inter-tree and intra-tree advantages.**
>
> Thanks for the question. As noted in `Sections 3.2` and `3.3`, we argue that the intra-tree advantage is more representative of the **symmetric step-level preference learning property**, while the inter-tree advantage can act as a **fallback baseline estimator when a single tree contains relatively few branches**.
>
> We extend the original `Table 4` to the following table here in order to better verify this viewpoint. When the rollout budget is relatively limited (`M=2,N=2,L=1`), each tree contains only three branches. In this case, the intra-tree advantage estimation exhibits high variance and the baseline estimation becomes unstable, leading to collapse in RL training. While when `N=6`, the intra-tree advantage can also conduct relatively stable RL training. In our experiments, combining intra-tree advantage with inter-tree advantage yields more stable and improved training performance across various `MNL` settings.
>
> | **Advantage Setting** (Qwen2.5-3b)                       | **Hotpot** | **2Wiki** | **Musiq** | **Bamb** | **Avg.** |
> | -------------------------------------------------------- | :--------: | :-------: | :-------: | :------: | :------: |
> | Chain-based ($M=4$)                                      |    39.0    |   36.3    |   15.2    |   36.8   |   31.8   |
> | $\hat{A}_\text{Intra-tree}$ ($M=2,N=2,L=1$)              |  collapse  | collapse  | collapse  | collapse | collapse |
> | $\hat{A}_\text{Inter-tree}$ ($M=2,N=2,L=1$)              |    40.6    |   41.3    |   16.5    |   36.8   |   33.8   |
> | $\hat{A}_\text{Intra-tree + Inter-tree}$ ($M=2,N=2,L=1$) |  **42.4**  | **43.7**  | **17.8**  | **43.2** | **36.8** |
> | Chain-based ($M=8$)                                      |    39.4    |   36.4    |   16.1    |   33.6   |   31.4   |
> | $\hat{A}_\text{Intra-tree}$ ($M=2,N=6,L=1$)              |    40.9    |   39.7    |   17.8    |   40.0   |   34.6   |
> | $\hat{A}_\text{Inter-tree}$ ($M=2,N=6,L=1$)              |    41.1    |   40.3    |   17.9    |   34.4   |   33.4   |
> | $\hat{A}_\text{Intra-tree + Inter-tree}$ ($M=2,N=6,L=1$) |  **42.2**  | **42.6**  | **18.3**  | **42.4** | **36.4** |
>
> The related part is updated in `Section 4.3 Tree-based Advantage`(line 461-472) and `Table 4`  of the revised version.

---

> ### Author Response · Authors · 2025-11-25
> **Looking Forward to Your Reply**
>
> Thank you once again for taking the time to review our paper and for providing constructive feedback.
>
> We have carefully considered each of your comments and have provided detailed responses. We sincerely hope that our efforts adequately address your concerns and contribute positively to your evaluation.
>
> As the author–reviewer discussion period progresses, we would greatly appreciate any further feedback you may have. If you have any additional questions or require any clarifications, please do not hesitate to reach out to us.

---

> ### Author Response · Authors · 2025-11-26
> **Official Comment by Authors (2/2)**
>
> > **W2&Q2: How to choose M,N,L in prior? Is M more imporntant than N?**
>
> Thanks for the valuable question. Based on the experiments and discussions in `Appendix B.4` and above `Q1`, we attribute that `M` governs the diversity of sampling (as **exploration**), whereas `N` and `L` govern the granularity of the process signal (as **exploitation**). And the granularity of process signals required for RL training is **strongly correlated with the interaction length of the task**, since the tree depth is directly dictated by the number of agent steps (*e.g.,* 3-4 steps for multi-hop QA). In this case, when `N` is of the same order of magnitude as or slightly larger than tree depth, it is generally sufficient to construct the necessary granularity for the process signals.
>
> Specifically, as shown in following table (results from `Table 3` and `Table 7` of the paper) for the multi-hop QA setting (tree depth around 4), when the rollout budget is small (*e.g.,* 4/per prompt), the choices for `M` and `NL` are very limited. When the rollout budget is relatively ample (*e.g.,* 16/per prompt), `N=5` is sufficient to construct process branch signals for each tree. Larger values of `N,L`(`N=11,L=1` and `N=6,L=2`) yield diminishing marginal benefits, while a relatively reduced `M` limits sampling diversity and leads to performance degradation.
>
> | **Tree Setting** (Qwen2.5-3b) | **Hotpot** | **2Wiki** | **Musiq** | **Bamb** | **Avg.** |
> | ----------------------------- | :--------: | :-------: | :-------: | :------: | :------: |
> | $M=4,N=0,L=0$                 |    39.0    |   36.3    |   15.2    |   36.8   |   31.8   |
> | $M=1,N=5,L=1$                 |    41.5    |   39.3    |   15.8    |   37.6   |   33.6   |
> | $M=2,N=1,L=2$                 |    42.3    |   43.2    |   17.6    |   41.9   |   36.3   |
> | $M=2,N=2,L=1$ (Best)            |  **42.4**  | **43.7**  | **17.8**  | **43.2** | **36.8** |
> | $M=16,N=0,L=0$                |    40.1    |   38.8    |   17.5    |   39.2   |   33.9   |
> | $M=8,N=2,L=1$                 |    44.6    |   43.2    |   18.2    |   38.4   |   36.1   |
> | $M=6,N=3,L=1$                 |  **45.3**  | **44.1**  | **18.8**  |   37.6   |   36.5   |
> | $M=4,N=5,L=1$  (Best)           |    45.0    |   43.9    |   18.5    | **41.6** | **37.3** |
> | $M=2,N=6,L=2$                 |    43.2    |   43.1    |   17.0    |   40.0   |   35.8   |
> | $M=2,N=11,L=1$                |    43.0    |   42.2    |   16.1    |   40.0   |   35.3   |
>
> In summary, when a sufficient rollout budget is available, choose `N,L` to values on the same order as the tree depth and allocating the remaining budget to `M` yields the best results.
>
> The related part is updated in `Appendix B.4` (line 1011-1020) of the revised version.
>
> ---
>
> > **W1: Marginal gains on web-agent benchmarks.**
>
> In this work, we attempt to employ a heuristic tree-structure sampling to construct process signal supervision, with the aim of achieving more effective credit assignment. Although credit assignment can improve the learning efficiency of RL, we believe that relatively simple and short-horizon agent training is hard to generalize well to more complex interactive tasks (like BrowseComp). Although some recent studies achieve relatively strong results on such challenging benchmarks, we think the primary improvement stems from scaling the agent environment and data construction. Our work and these works originate from different perspectives, and we believe they are all valuable for addressing and optimizing agent RL. We are also currently working on constructing more complex training data for enhancement.
>
> ---
>
> We sincerely appreciate your feedback. Hope the above response can address your concerns. We are pleased to provide further clarification if you have any further questions!

---

> ### Author Response · Authors · 2025-11-27
> **Kindly Reminder for Your Feedback**
>
> Dear Reviewer PcpF,
>
> We appreciate your time and effort to review our work. We have carefully considered each of your comments and done our utmost to address your concerns.
>
> As the rebuttal period draws to a close, we would like to kindly remind that if you could review our responses and let us know whether it address your points, or if there is any additional clarification needed.
>
> Best,
>
> The authors of Paper 1473

---

> > ### Comment · Reviewer_PcpF · 2025-11-28
> >
> > Thanks for your response. You clearly explain how to select M, N, L based on the task. However, the motivation to adopt Tree-GRPO is that it can handle more complicated tasks, it shows that the performance on the complicated benchmarks (web-agent) is only marginal improvement. I decide to maintain the score.

---

> ### Author Response · Authors · 2025-11-28
> **Thanks for The Support**
>
> Thanks for your reply. We truly appreciate your continued engagement during the rebuttal process, which helps us improve the manuscript. As you suggested, we will further validate the effectiveness of Tree-GRPO in future work by scaling both the environment and complexity of the data.

---

### Author Response · Authors · 2025-11-21
**Response and Updated Manuscript to All Reviewers**

We sincerely thank all the reviewers for their thoughtful and constructive feedback, which has been valuable in improving our work. In particular, we appreciate the specific acknowledgement from reviewers:

- `PcpF` and `uD4w` for valuing theoretical justification that connects Tree-GRPO to well-established step-level DPO
- `PcpF`,`TDPe` and `uD4w` for highlighting the paper exceptional quality, both on well-written and insightful experiments
- `TDPe` and `uD4w` for recognizing that agent thought-action-observation step-level node is promising and clever

Based on the valuable feedback, we try to address all concerns by comments and updating our manuscript. Updates are shown in orange for clarity, and below is the summary of changes:

- More experiments and discussion on $\hat{A}_\text{Intra-tree/Inter-tree}$ and $M,N,L$ choices (by `PcpF`)
- More thoughts and discussion about existing sparse-reward RL (by `TDPe`)
- More ablations on KL term (by `uD4w`)

We believe these updates further justify our approach to addressing critical challenges in LLM agent RL. We hope the revised submission meets your expectations and demonstrates the value of our contributions.

Thanks again for your constructive feedback and support!

---

### Author Response · Authors · 2025-11-26
**Gentle Follow-up as Discussion Nearing Close**

Dear Reviewer `PcpF` and `TDPe`,

A gentle follow-up as the discussion period is nearing its conclusion. We have posted all the responses and are fully available to address any remaining questions or concerns you may have.

We welcome any further feedback you may have. Thank you for your valuable time and consideration.

Best,

The authors of Paper 1473

---

### Author Response · Authors · 2025-12-02
**Summary of Reviews and Rebuttals**

Dear ICLR 2026 AC, SAC and PC,

First we would like to sincerely thank everyone involved in the review process for the time and effort in evaluating our work. Here is a summary of the review and rebuttal content for easier reference.

---
### **Summary of Reviews**

The paper received a total of three initial reviews, with **PcpF** and **uD4w** both giving an overall rating of **6**, and **TDPe** gives a **4**. Following the rebuttal, both **PcpF** and **uD4w** confirm their major concerns are addressed and decide to maintain score at **6**, whereas no any further feedback from **TDPe** are received. Specifically:

- Reviewer **PcpF** (rating: 6) raised questions regarding the tree-based advantages, and the choice of tree‑search parameters $M,N,L$. After our response and revised manuscript, **PcpF** commented: `clearly explain how to select M, N, L based on the task`, and `decide to maintain the score`.
- Reviewer **uD4w** (rating: 6) requested comparisons of actual budget savings and wall‑clock time, as well as additional ablation studies. After our response and revised manuscript, **uD4w** stated: `have addressed my major concerns` and `glad to see the extended ablation study`.
- Reviewer **TDPe** (rating: 4) has two questions. (i) The first is identical stepwise advantages may fail to capture varied contribution (which is a **misunderstanding**) and requests for justification. We responded with a simple illustrative example and further analysis to justify and easier understanding, along with comparisons with other credit assignment methods. (ii) The second is asking for providing thoughts on existing sparse-reward RL on LLM agent. As suggestion, we discuss existing sparse‑reward RL into three categories, and further provide thoughts of applying them on LLM agent era. We believe that all concerns have been addressed. **Unfortunately, we did not receive any further feedback or engagement**.

---
### **Summary of Rebuttals**
In the rebuttal period, we address all concerns in detail, and the corresponding parts are updated in the revised version. In summary:

|Reviewers|Concerns/Questions|Our Responses and Actions|
|--|--|--|
|**PcpF**|$\hat{A}\_\text{Intra-tree/Inter-tree}$|Extend experiments on adv ablation on different tree structures, clarify two adv contributions. Updated on `Sec 4.3 Tree-based Advantage`.|
| | How to choose $M,N,L$ in prior?|Provide further discussion on different tree settings, explain how $M,N,L$ contribute for specific tasks and budgets. Updated on `Appendix B.4`.|
|**uD4w**|Practical details of rollout savings|Provide practical rollout savings, and show statistical results for 1.5× sample num with the same budget (as stated in paper).|
| |Actual wall-clock time comparison|Provide actual wall-clock time comparison between tree/chain-based methods, and indicate actual limitations.|
| |Comparison with explicit PRM|List current PRMs and their limitations, and compare with recent results of explicit process supervision methods.|
| |Ablation on KL term|Extend ablation study on KL term, demonstrate robustness of Tree-GRPO. Updated on `Sec 4.3 Chain-based vs. Tree-based` and `Fig 6`.|
|**TDPe**|Identical credits can't capture step contribution|Justify the **misunderstanding** through analysis and examples, and compare different step‑based credit assignment methods (formally and experimentally). |
| |Provide thoughts on existing sparse-reward RL|Discuss existing sparse‑reward RL across three categories, and propose corresponding thoughts. Compare with 4 other sparse-reward RL methods for LLMs. Updated on `Sec 5 Related Work`.|
---
### **Major Contributions and Stengths**

This work proposes Tree-GRPO, a tree‑based grouped relative policy optimization designed to address two key challenges in agent RL: *high sampling cost* and *sparse supervision from outcome rewards*. The major contributions and acknowledged strengths from reviews are highlighted:
- **Promising/Original Idea:** Represent each agent t–a–o step as a tree node is a promising (**TDPe**) and clever adaptation (**uD4w**), which enables more rollouts under the same budget (**PcpF**).
- **Sound Theory:** Not only propose a method, but also provide theoretical justification that connects Tree-GRPO to well-established step-level DPO (**PcpF**, **uD4w**).
- **Insightful Analysis:** The experiments are extensive (**TDPe**) and thorough (**uD4w**), and the quality is further reinforced by insightful ablation studies (**PcpF**).
- **High Quality:** The overall quality is exceptional (**uD4w**, **PcpF**, **TDPe**), both on well written and outstanding clarity. Offers substantial significance to the field (**uD4w**).
- **Superior Performance:** Superior results across 11 datasets and 3 types of agent tasks. Enable better performance under a lower rollout budget (*e.g.*, 1/4).
---

We hope this summary of reviews and rebuttals could bring a quick understanding of the reviewers’ evaluations and comments.

Best,

The authors of paper 1473

---

### Meta-Review · Area_Chair_DonJ · 2025-12-16

**Summary:**

Overall, the reviews were medium to positive. The authors made significant efforts to address the reviewer concerns, through significant paper updates, additional experiments, and detailed clarifications. Therefore, I would expect at least some of the reviewers would have raised their scores, resulting in a consensus to accept the paper. Therefore, my recommendation is to **accept**. Details are below.

Reviewer PcpF recommended borderline accept. They raised two concerns. First, they brought up as a weakness that the proposed method did not succeed on the hardest task (web agent benchmarks).  Second, they noted that the proposed method introduces three new hyperparameters (M, N, L), and were concerned that these might require significant tuning. In their response, the authors conducted an ablation study on the additional hyperparameters and gave guidelines how to set them based on compute budget. I think this at least partially addresses the reviewers concern. I do not think their concern about the poor performance on web agent benchmarks is fundamental given that the baselines also performed poorly there. I would expect this reviewer to raise their score from a 6 to an 8.

Reviewer uD4w recommended borderline accept. They listed concerns around a lack of ablations of the KL coefficient, one of the assumptions in the theoretical justification being too restrictive, some missing process reward baselines, and detailed wall clock comparisons. In their response, the authors added experiments or clarifications concerning all of these points. Therefore, I would expect the reviewer to raise their score from 6 to 8.

Reviewer TDPe recommended borderline reject. They listed as a weakness that this paper does not compare to classic RL methods for dealing with sparse rewards (novelty bonuses, hindsight experience replay, and the like). In response, the authors have updated their related work section to include a discussion of these methods. I think this is satisfactory, since the proposed method is orthogonal and could in principle be combined with these methods. They also raised a question about how credits were distributed across different nodes in the tree (i.e. depth vs. breadth). The authors have provided what I think is a satisfactory response, by giving simple examples how credits are same within the same branch, but differ across branches. I would expect this reviewer to raise their score from a 4 to 6.

**Reviewer Concerns:**

I think most of the reviewer concerns have been addressed by the author responses and paper updates.

**Reviewer Scores:**

PcpF: 6 -> 8

uD4w: 6 -> 8

TDPe: 4 -> 6

---

### Decision · Program_Chairs · 2026-01-26

Accept (Poster)